

# Aluminous clay and pedogenic Fe oxides modulate aggregation and
# related carbon contents in soils of the humid tropics
Maximilian Kirsten*[1], Robert Mikutta[2], Didas N. Kimaro[3], Karl-Heinz Feger[1], Karsten Kalbitz[1]
[1] Technische Universität Dresden, Institute of Soil Science and Site Ecology, Tharandt, Germany
[2] Martin-Luther-Universität Halle-Wittenberg, Soil Science and Soil Protection, Halle/Saale Germany
[3] Mwenge Catholic University, Directorate of Research Innovations and Consultancy, Moshi, Tanzania

22  **Keywords**: tropical soils, aggregate size distribution, aggregate stability, soil mineralogy, kaolinite,

23  gibbsite, goethite, hematite, land-use change

* maximilian.kirsten@tu-dresden.de



## Abstract

Aggregation affects a wide range of physical and biogeochemical soil properties with positive feedbacks on soil carbon storage. For weathered tropical soils, aluminous clays (kaolinite and gibbsite) and pedogenic Fe (oxyhydr)oxides (goethite and hematite; termed 'Fe oxides') have been suggested as important building units for aggregates. However, as both secondary aluminosilicates and Fe oxides are part of the clay-sized fraction it is hard to separate, how certain mineral phases modulate aggregation, and what consequences this has for organic carbon (OC) persistence after land-use change. We selected topsoils with unique mineralogical compositions in the East Usambara Mountains of Tanzania under forest and cropland. Soils are varying in contents of aluminous clay and Fe oxides. Across the mineralogical combinations, we determined the aggregate size distribution, aggregate stability, OC contents of aggregate size fractions as well as changes in aggregation and OC contents under forest and cropland land use. We found the soil aggregation patterns (high level of macroaggregation and aggregate stability) more similar than different among mineralogical combinations. Yet, an aluminous clay content $> 250$ g kg$^{-1}$ in combination with pedogenic Fe contents $< 60$ g kg$^{-1}$ significantly promoted the formation of large macroaggregates $> 4$ mm. In contrast, a pedogenic Fe content $> 60$ g kg$^{-1}$ in combination with aluminous clay content of $< 250$ g kg$^{-1}$ promoted OC storage and persistence after the change in land use. The low clay-high Fe combination displayed the highest OC persistence, despite conversion of forest to cropland caused substantial disaggregation. Our data indicate that aggregation in this typical soil of the humid tropics is modulated by the mineralogical regime, causing moderate but significant differences in aggregate size distribution. Nevertheless, aggregation was little decisive for overall OC persistence in the highly weathered soils, where OC storage is more regulated by direct mineral-organic interactions.





## 1. Introduction

Many functions of soils such as food production, water purification as well as climate regulation are tightly linked to soil structure (*Bronick* and *Lal*, 2005; *FAO*, 2015; *Six* et al., 2004). Aggregates are the structural backbone of soil and changes in aggregation impacts various processes such as root development, soil erosion, and soil organic carbon (OC) accumulation (*Chaplot* et al., 2010; *Le Bissonnais* et al., 2018). Based on their size, soil aggregates are typically classified into small microaggregates (< 20 µm), large microaggregates (20–250 µm), and macroaggregates (> 0.25 mm) (*Tisdall* and *Oades*, 1982). Cementing agents such as clay minerals, metal (oxyhydr)oxides, as well as organic matter (OM) are considered as primary building units of microaggregates (*Totsche* et al., 2018), which provide the basis for the formation of larger soil structural units (*Asano* and *Wagai*, 2014). Especially in weathered tropical soils, aggregation depends strongly on inorganic cementing agents (*Six* et al., 2002). Pedogenic iron (Fe) (oxyhydr)oxides (summarized as 'Fe oxides') have been reported to facilitate macroaggregation (*Peng* et al., 2015) and aggregate stability (*Duiker* et al., 2003). Under the acidic conditions of weathered tropical soils, Fe oxides provide positively charged surfaces capable of reacting with negatively charged inorganic constituents, like clay minerals or OM (*Kaiser* and *Guggenberger*, 2003; *Kleber* et al., 2015; *Six* et al., 2004; *Totsche* et al., 2018). At present, however, there is little consensus to which extent aggregation can be ascribed to individual inorganic or organic cementing agents, or whether aggregation is best explained by their mutual interactions. For example, the extent of aggregation has been either positively related to the contents of clay and OC (*Chaplot* and *Cooper*, 2015; *Paul* et al., 2008; *Spaccini* et al., 2001), or to differences in the clay mineral composition (*Fernández-Ugalde* et al., 2013). Furthermore, *Barthès* et al. (2008) showed that texture had no effect on macroaggregation over a range of tropical soils characterized by low-activity clay minerals. Uncertainty also derives from the fact that the clay size particle fraction (< 2-µm) not only contains OM and different types of clay minerals, but also variable contents of pedogenic Fe and aluminum (Al) oxides (*Barré* et al. 2014; *Fernández-Ugalde* et al. 2013; *Wagai* and *Mayer* 2007). *Denef* et al. (2004) showed that significant differences in the amount of microaggregates encased in macroaggregates can be related to the clay mineral composition (2:1, mixed layer, 1:1 clays).



They assume that interactions of 1:1 clay minerals with Fe oxides cause a higher aggregate stability
compared to those involving 2:1 clay minerals (*Denef* et al., 2002, 2004). Such mutual interactions
between typical aluminous clay-sized minerals (e.g. kaolinite, gibbsite) and pedogenic Fe oxides are thus
possible drivers of aggregation in weathered tropical soils (*Durn* et al., 2019).

As indicated above, soil aggregation is considered to be an important process that increases OC

persistence, because of the physical separation of OM from microorganisms and their exoenzymes (*Six* et
al., 2004). Thus, improved aggregation could contribute to enhanced OC storage in soils (*Kravchenko* et
al., 2015; *Marín-Spiotta* et al., 2008; *Schmidt* et al., 2011). Managing aggregation, *e.g.*, for climate change
mitigation, requires profound knowledge on the controls of aggregation and their effects on OC
persistence (*Paul* et al., 2008). To the best of our knowledge there are no studies available, which
investigated the influence of changes in the content of clay minerals with low activity and the content of
pedogenic metal oxides on aggregation under comparable mineralogical conditions for weathered tropical
soils. Macroaggregates are particularly susceptible to soil management (*Six* et al., 2000a; *Totsche* et al.,
2018). Consequently, destruction of macroaggregates upon changes from forests to cropland might
account for OC losses that were observed in tropical soils (e.g. *Don* et al., 2011; *Kirsten* et al., 2019;
*Mujuru* et al., 2013). The stability of aggregates should thus determine OC losses induced by land-use
change and higher losses should be related to lower aggregate stability (*Denef* et al., 2002; *Le Bissonnais*
et al., 2018; *Six* et al., 2000b). At present, we are not aware of any studies resolving the puzzle to which
extent aluminous clay and pedogenic Fe oxides, control soil aggregation as well as OC storage in
weathered tropical soils.

This study takes advantage of unique mineralogical combinations of soils in the East Usambara

Mountains of Tanzania. The soils vary in the amount of aluminous clay (kaolinite, gibbsite) and pedogenic
Fe oxides (goethite, hematite) but without variation in their mineralogical composition (*Kirsten* et al.,
2021). The small-scale conversion of natural forest to cropland that took place in that region enables us to
evaluate the effect of land-use change under each mineralogical combination on soil physical properties
and related OC persistence. In detail, our main research objective was to investigate into the individual



role of aluminous clay and pedogenic Fe oxides for determining (i) the soil aggregate size distribution, (ii)
aggregate stability, (iii) the consequences for OC allocation into different aggregate size fractions, and (iv)
the consequences for OC persistence related to land-use change. We presume that the mineralogical
combination resulting in the largest aggregate stability also results in largest OC persistence after
conversion of forests into croplands. Since land use induced OC losses in this region largely occur in
topsoils (*Kirsten* et al., 2019), we concentrated on samples from that part of the soil.



## 2. Material and methods

### 2.1 Study area and soil sampling

The study was conducted in the Eastern Usambara Mountains of Tanzania close to the village Amani (5°06′00″ S; 38°38′00″ E). The climate is humid monsoonal with a mean annual precipitation of 1,918 mm, and a mean annual temperature of 20.6°C with low variability within the study area (*Hamilton* and *Bensted-Smith*, 1989). The dominating Acrisols and Alisols, developed from Precambrian crystalline bedrock, are deeply weathered and highly leached, with visible clay illuviation in the subsoil (*Kirsten* et al., 2019). Criteria for site selection and soil sampling has been described in detail by *Kirsten* et al. (2021). Briefly, all soil samples were collected on mid-slope position. We sampled six plots under forest and three under annual cropping. Soil from three adjacent and randomly distributed soil pits was sampled at 0–5 and 5–10 cm depth. Living roots were removed and aliquots of the soils were sieved to < 2 mm after drying at 40°C. For each depth increment, three undisturbed soil cores (100 cm$^3$) were collected for bulk density determination.

### 2.2 Soil analyses

*Basic soil properties and selected mineralogical combinations*

Bulk density was determined after drying the soil at 105°C and corrected for coarse fragments (*Carter* and *Gregorich*, 2008). Soil pH was measured in 0.01 M CaCl$_2$ at a soil to solution ratio of 1 : 2.5. Extraction of poorly crystalline Fe and Al phases as well as of Fe and Al complexed by OM was done with ammonium oxalate according to *Schwertmann* (1964). Effective cation exchange capacity (CEC$_{eff}$) and base saturation (BS) were determined following the procedure provided by *Trüby* and *Aldinger* (1989). Contents of OC and total N were analyzed by high temperature combustion at 950°C and thermo-conductivity detection (Vario EL III/Elementar, Heraeus, Langenselbold, Germany). A combined dithionite-citrate-bicarbonate extraction and subsequent texture analysis was applied to determine the contents of aluminous clay and total pedogenic Fe (Fe$_d$). Details of the procedure are described in *Kirsten* et al. (2021). Based on the 5–10 cm depth increment, we differentiated four groups varying in contents of



aluminous clay and pedogenic Fe oxides under forest (i.e. ʻlow clay–low Fe', ʻlow clay–high Fe', ʻhigh
clay–low Fe', ʻhigh clay–high Fe'), and three analogous groups under cropland (i.e. ʻlow clay–low Fe',
ʻlow clay–high Fe', ʻhigh clay–high Fe').

*Aggregate size distribution, aggregate stability and carbon contents*
Aggregate size distribution was determined by dry sieving as it most closely resembles soil conditions at
the end of the long dry season. Undisturbed soil was dried at 40°C for 48 hours. Separation of aggregate
sizes was conducted with a sieving machine (AS 200 control "g", Retsch, Hanau, Germany) combined
with a set of four sieves with meshes of 4, 2, 1, and 0.25 mm, respectively (*Larney*, 2008). The amplitude
was set to 1.51 mm (7.6 *g*-force), which was applied over a sieving duration of three minutes. Aggregate
stability was tested for the two largest aggregate size fractions (2–4 mm and > 4 mm). The fast wetting
pretreatment was applied to both fractions (*Le Bissonnais*, 1996) using a wet-sieving apparatus
(Eijkelkamp, Giesbeek, Netherlands) with sieve openings of 63 μm. This procedure simulates the
transition of aggregates from dry to rainy season. Sieving was conducted in ethanol for three minutes
(stroke 1.3 cm, f = 34 min$^{-1}$). All aggregates remaining on the sieve were dried at 105°C. Water-stable
aggregates were subsequently introduced to a sieving apparatus with a set of five sieves with mesh sizes of
4, 2, 1, 0.63, and 0.25 mm, respectively (*Larney*, 2008). For each obtained aggregate fraction by dry
sieving, OC contents analyzed by high temperature combustion at 950°C and thermo-conductivity
detection (Vario EL III/Elementar, Heraeus, Langenselbold, Germany). The mass corrected OC content of
a certain aggregate fraction was calculated using equation 1 to resemble the contribution to total soil OC,
$Mass-corrected\ OC_{Aggregate} = \frac{m_i}{\sum_{i=0}^{n} m_i} \times OC_{Aggregate}$ (Eq. 1)
where $m_i$ represents the mass of an aggregate size fraction (g), $\sum m_i$, the sum of masses of all size
fractions (g), and $OC_{Aggregate}$ the OC content of aggregate fraction "$i$".
The mean weight diameter (MWD) of aggregates was calculated using equation 2 for undisturbed soil to
describe the initial aggregate size distribution, and for the large aggregate size fractions after exposure to
the stability test to evaluate the effect of fast wetting on aggregate stability,





$$MWD = \sum_{i=0}^{n} \frac{m_i}{\sum m_i} \times d_i \qquad \text{(Eq. 2)}$$
where $m_i$ represents the mass of an aggregate size fraction (g), $\sum m_i$, the sum of masses of all size
fractions (g), and $d_i$ the mean mesh diameter of fraction "$i$" (mm). The MWD of the aggregate fraction
> 4 mm was estimated by doubling the largest sieve size diameter (*Youker* and *McGuinness*, 1957).

**2.3 Statistics and calculations**
The mean and standard deviation of data were calculated with the software package R (version 3.6.0). To
test for significant differences between treatments, linear model function [lm()] was used in combination
with analysis of variance [aov(lm()]. The Tukey-HSD test was used as a post-hoc comparison of means;
the LSD-test was applied in the case of non-equality of variances. Linear regression and correlation
analysis was used to test for relations between independent variables. Statistical differences are reported at
a significance level of $p < 0.05$.



## 3. Results

### 3.1 Mineralogical composition and general soil properties

The mineralogical composition of the study soils was very homogeneous with kaolinite and gibbsite as the main aluminous minerals of the clay fraction, and well-crystalline goethite and hematite as dominant pedogenic Fe oxides (cf. *Kirsten* et al., 2021). The selected mineralogical combinations represent a broad spectrum of possible combinations in both mineral constituents. Amounts of aluminous clay varied between 149 and 438 g kg$^{-1}$, and Fe$_d$ between 21 and 101 g kg$^{-1}$ across all sites and land uses. Amorphous Fe and Al phases contributed little to pedogenic oxides as indicated by low proportions of oxalate-extractable Fe and Al (Table 1). The advanced weathering state of study soils was also reflected in low pH and CEC$_{eff}$ values (Table 1; *Kirsten* et al., 2021).



**Table 1**: Basic properties of the two soil depth increments sampled along the mineralogical combinations with aluminous clay (clay), dithionite-citrate-bicarbonate-extractable Fe ($Fe_d$), effective cation exchange capacity ($CEC_{eff}$), total soil organic carbon content (OC), $Fe_d$ to aluminous clay ratios ($Fe_d$/clay), hydrogen peroxide- and dithionite-citrate-bicarbonate-treated sand and silt contents, and oxalate-extractable Fe and Al contents ($Fe_o$ and $Al_o$). Aluminous clay represents the weight sum of kaolinite and gibbsite present in the $< 2$-µm fraction after removal of OM and pedogenic Fe oxides. Lower case letters indicate significant differences within a certain land use as separated by depth. Sample numbers for the combinations are as follows: 'low clay–low Fe' under forest ($n = 4$), 'low clay–high Fe' under forest ($n = 4$), 'high clay–low Fe' under forest ($n = 4$), 'high clay–high Fe' under forest ($n = 3$), 'high clay–high Fe' under forest ($n = 3$); all cropland combinations ($n = 3$).

| Land use | Mineralogical Combination | Depth (cm) | Sand (g kg$^{-1}$) | Silt (g kg$^{-1}$) | Clay (g kg$^{-1}$) | $Fe_d$ | $Fe_d$/clay | $Fe_o$ (g kg$^{-1}$) | $Al_o$ (g kg$^{-1}$) | OC (g kg$^{-1}$) | pH (0.01 M CaCl$_2$) | $CEC_{eff}$ (cmol$_c$ kg$^{-1}$) |
|---|---|---|---|---|---|---|---|---|---|---|---|---|
| Forest | **Low** aluminous clay– **Low** pedogenic Fe oxides | 0–5 | 788[a] (21) | 63[c] (24) | 149[b] (19) | 21[d] (4) | 0.15[b,A] (0.04) | 1.4[a] (0.3) | 1.2[a] (0.2) | 76[ab,A] (27) | 3.5[b] (0.1) | 5.7[a] (2.6) |
| | | 5–10 | 712[a] (46) | 107[b] (57) | 181[b] (19) | 38[b] (13) | 0.21[bc,A] (0.09) | 1.8[a] (0.3) | 1.4[a] (0.2) | 34[a,A] (6) | 3.7[b] (0.1) | 2.9[a] (0.1) |
| Forest | **Low** aluminous clay– **High** pedogenic Fe oxides | 0–5 | 617[b] (36) | 201[a] (52) | 182[b] (38) | 78[a] (14) | 0.45[a,A] (0.12) | 1.3[a] (0.2) | 1.5[a] (0.2) | 57[b,A] (14) | 3.8[a] (0.2) | 5.6[a] (1.7) |
| | | 5–10 | 647[b] (49) | 179[a] (26) | 174[b] (42) | 77[a] (4) | 047[a,A] (0.13) | 1.3[b] (0.1) | 1.6[a] (0.3) | 37[a,A] (7) | 3.8[ab] (0.1) | 3.2[a] (0.9) |
| Forest | **High** aluminous clay– **Low** pedogenic Fe oxides | 0–5 | 571[c] (19) | 131[b] (32) | 298[a] (41) | 36[c] (5) | 0.12[b] (0.01) | 0.9[b] (0.0) | 1.3[a] (0.2) | 43[b] (6) | 4.0[a] (0.2) | 5.2[a] (1.1) |
| | | 5–10 | 489[c] (24) | 137[ab] (1) | 374[a] (24) | 44[b] (7) | 0.12[c] (0.02) | 1.0[b] (0.1) | 1.5[a] (0.3) | 23[b] (5) | 3.9[ab] (0.1) | 3.0[a] (0.4) |
| Forest | **High** aluminous clay– **High** pedogenic Fe oxides | 0–5 | 530[c] (28) | 152[b] (24) | 318[a] (41) | 67[b] (5) | 0.22[b,A] (0.03) | 1.2[ab] (0.3) | 1.9[a] (0.8) | 95[a,A] (31) | 4.1[a] (0.2) | 7.8[a] (1.8) |
| | | 5–10 | 473[c] (35) | 178[a] (45) | 349[a] (40) | 81[a] (6) | 0.23[a,A] (0.02) | 1.3[b] (0.1) | 1.7[a] (0.2) | 35[a,A] (5) | 4.0[a] (0.1) | 4.9[a] (4.0) |
| Cropland | **Low** aluminous clay– **Low** pedogenic Fe oxides | 0–5 | 670[a] (8) | 103[c] (4) | 227[b] (6) | 30[c] (2) | 0.13[b,A] (0.01) | 0.6[c] (0.0) | 1.1[a] (0.1) | 19[c,B] (0) | 5.0[b] (0.1) | 5.1[b] (0.2) |
| | | 5–10 | 669[a] (8) | 118[b] (28) | 213[b] (24) | 29[c] (4) | 0.14[b,A] (0.03) | 0.6[c] (0.0) | 1.1[a] (0.1) | 19[c,B] (1) | 5.0[ab] (0.1) | 5.1[b] (0.2) |
| Cropland | **Low** aluminous clay– | 0–5 | 602[b] (17) | 200[a] (13) | 198[b] (29) | 101[a] (4) | 0.51[a,A] (0.0) | 1.5[a] (0.0) | 4.1[a] (0.2) | 47[a,A] (1) | 4.9[c] (0.1) | 5.1[b] (0.2) |



|  |  |  |  |  |  |  |  |  |  |  |  |
|---|---|---|---|---|---|---|---|---|---|---|---|
|  | **High** pedogenic Fe oxides | 5–10 | **579**[b] (19) | **206**[a] (4) | **215**[b] (23) | **100**[a] (5) | **0.47**[a, A] (0.07) | **1.7**[a] (0.1) | **4.3**[a] (0.6) | **48**[a,A] (5) | **4.8**[b] (0.1) | **5.0**[b] (1.2) |
| **Cropland** | **High** aluminous clay– | 0–5 | **437**[c] (14) | **129**[b] (12) | **434**[a] (18) | **63**[b] (3) | **0.15**[b,B] (0.01) | **1.2**[b] (0.0) | **1.4**[b] (0.0) | **34**[b,B] (1) | **5.4**[a] (0.0) | **9.4**[a] (0.5) |
|  | **High** pedogenic Fe oxides | 5–10 | **399**[c] (18) | **163**[ab] (35) | **438**[a] (17) | **66**[b] (4) | **0.15**[b,B] (0.01) | **1.2**[b] (0.1) | **1.3**[b] (0.2) | **30**[b,A] (3) | **5.2**[a] (0.1) | **7.3**[a] (0.7) |

### 3.2 Aggregate size distribution

The studied soils were highly aggregated and showed significant variation in their aggregate size distribution across the mineralogical combinations (Table 2). For the low clay–low Fe combination under forest, about 40% of the total soil mass prevailed in $> 2$ mm aggregates, while in the high clay–low Fe combination 74% were assigned to this fraction (Figure 1a). Furthermore, only 3–12% of total soil mass remained in $< 0.25$ mm aggregates (Table 2). The low clay–low Fe combination under forest displayed the significant smallest MWD, with 2.9 mm in 0–5 cm depth and 3.7 mm in 5–10 cm depth (Table 2). In contrast, the low clay–high Fe combination always had the largest MWD (4.8 mm in 0–5 cm depth, and 4.6 mm in 5–10 cm depth) among the other forest combinations. Our data suggest that the MWD under forest is significantly positively influenced by the $Fe_d$ content ($MWD_{Forest\ 0–5\ cm}$: $r^2 = 0.4$, $p < 0.001$; $MWD_{Forest\ 5–10\ cm}$: $r^2 = 0.15$, $p = 0.06$), whereas nearly no effect was observed for aluminous clay ($MWD_{Forest\ 0–5\ cm}$: $r^2 < 0.01$, $p = 0.79$; $MWD_{Forest\ 5–10\ cm}$: $r^2 < 0.01$, $p = 0.30$, Table S1). Contrary to the mineralogical combinations under forest, the significant smallest MWD under cropland was within the low clay–high Fe combination (2.7 mm in 0–5 cm depth and 2.7 mm in 5–10 cm depth; Table 2). The low clay–low Fe and high clay–high Fe cropland combinations showed no strong differences in their MWDs. Nonetheless, a significant negative linear relationship existed between MWD and the pedogenic-Fe to aluminous clay ratio ($MWD_{Cropland\ 0–5\ cm}$: $r^2 = 0.47$, $p = 0.03$; $MWD_{Forest\ 5–10\ cm}$: $r^2 = 0.47$, $p = 0.02$) for the mineralogical combinations under cropland (Table S1).

Corresponding to the smallest MWD, the low clay–low Fe forest combination contained the smallest fraction of $> 4$ mm aggregates. The contribution of these large aggregates under forest increased in the order: low clay–low Fe < low clay–high Fe = high clay–high Fe < high clay–low Fe (Figure 1a). For croplands, the low clay–high Fe combination comprised the smallest amount of $> 4$ mm aggregates whereas the high clay–high Fe combination exhibited the respective highest share (Figure 1a). The explained variance of $> 4$ mm aggregate mass due to aluminous clay and $Fe_d$ was generally low, except for the cropland combinations (positive effect of aluminous clay and negative effect of pedogenic Fe; Table S1).



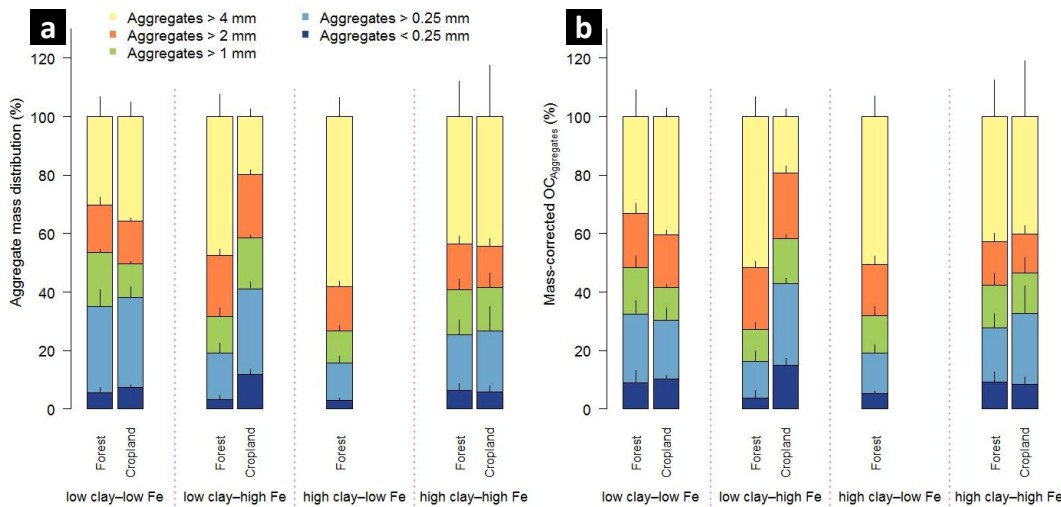

**Figure 1**: Aggregate size distribution of the combined 0–5 and 5–10 cm depth increments (a), and relative mass-corrected OC contents (b) along the mineralogical combinations. Clay represents the weight sum of kaolinite and gibbsite present in the < 2-µm fraction after removal of OM and pedogenic Fe oxides, and Fe denotes the content of pedogenic Fe oxides extracted with dithionite-citrate-bicarbonate. Sample numbers for the combinations are as follows: 'low clay–low Fe' under forest ($n = 4$), 'low clay–high Fe' under forest ($n = 4$), 'high clay–low Fe' under forest ($n = 3$), 'high clay–high Fe' under forest ($n = 7$); all cropland combinations ($n = 3$).

The mineralogical combinations affected the amounts of 2–4 mm aggregates differently than those of > 4 mm aggregates. The low clay–high Fe combination under forest and cropland contained slightly but significantly more 2–4 mm aggregates (Figure 1a), being associated with a significantly higher $Fe_d$ to aluminous clay ratio (Table 1). In fact, in a multiple regression model for the entire data set (combined land uses and depths), we observed a positive relationship between the mass of 2–4 mm aggregates and $Fe_d$ content, whereas the content of aluminous clay had a negative effect ($r^2 = 0.57$, p < 0.001; Table S1). The same model separated by soil depth showed similar relationships (Table S1). Across all mineralogical combinations, amounts of < 0.25 mm aggregates were principally comparable, despite of significantly higher shares in the low clay–low Fe and high clay–high Fe combinations under forest. In contrast, a significant larger amount of < 0.25 mm aggregates was observed in the low clay–high Fe combination under cropland. In this mineralogical combination, land-use change caused a quadrupling of < 0.25 mm aggregate mass from about 30 to nearly 120 g kg$^{-1}$ (Table 2). In contrast to the macroaggregate fractions



shown above, there was no correlation between mineralogical parameters and the mass of $< 0.25$ mm
aggregates, neither for the entire data set (combined land uses and depths) nor when separated by soil
depth (Table S1). Only under cropland we observed a negative effect of aluminous clay and a positive
influence of $Fe_d$ on microaggregate contents (aggregate mass $< 0.25$ $mm_{0-5\ cm}$: $r^2 = 0.8$, $p = 0.004$;
aggregate mass $< 0.25$ $mm_{5-10\ cm}$: $r^2 = 0.61$, $p = 0.03$).





**Table 2**: Aggregate masses (mass) and OC content of aggregate size fractions (dry sieving) within different combinations of aluminous clay and
pedogenic Fe oxides, OC change (ΔOC) between land uses within a certain mineralogical combination and depth, and related mean weight diameter
(MWD). Aluminous clay represents the weight sum of kaolinite and gibbsite present in the < 2-µm fraction after removal of OM and pedogenic Fe
oxides. Lower case letters indicate significant differences within a certain land use separated by depth, and capital letters denote significant differences
between land uses. Sample numbers for the combinations are as follows: 'low clay–low Fe' under forest ($n = 4$), 'low clay–high Fe' under forest ($n =$
4), 'high clay–low Fe' under forest ($n = 3$), 'high clay–high Fe' under forest ($n = 7$); all cropland combinations ($n = 3$).

| Land use | Mineralogical Combination | Depth (cm) | > 4 mm mass (g kg⁻¹) | > 4 mm OC (g kg⁻¹) | > 4 mm ΔOC (%) | 2–4 mm mass (g kg⁻¹) | 2–4 mm OC (g kg⁻¹) | 2–4 mm ΔOC (%) | 1–2 mm mass (g kg⁻¹) | 1–2 mm OC (g kg⁻¹) | 1–2 mm ΔOC (%) | 0.25–1 mm mass (g kg⁻¹) | 0.25–1 mm OC (g kg⁻¹) | 0.25–1 mm ΔOC (%) | < 0.25 mm mass (g kg⁻¹) | < 0.25 mm OC (g kg⁻¹) | < 0.25 mm ΔOC (%) | MWD (mm) |
|---|---|---|---|---|---|---|---|---|---|---|---|---|---|---|---|---|---|---|
| Forest | **Low aluminous clay–** | 0–5 | 249$^{c,A}$ (33) | 76$^{a,A}$ (32) | na | 144$^{b,A}$ (21) | 83$^{a,A}$ (22) | na | 191$^{a,A}$ (4) | 65$^{a,A}$ (9) | na | 345$^{a,A}$ (40) | 56$^{a,A}$ (18) | na | 70$^{a,A}$ (15) | 125$^{ab,A}$ (51) | na | 2.9$^{c,A}$ (0.3) |
| | **Low pedogenic Fe oxides** | 5–10 | 343$^{b,A}$ (61) | 40$^{a,A}$ (8) | na | 176$^{ab,A}$ (21) | 39$^{a,A}$ (10) | na | 181$^{a,A}$ (15) | 27$^{a,A}$ (9) | na | 257$^{a,A}$ (36) | 28$^{a,A}$ (5) | na | 44$^{a,B}$ (11) | 51$^{a,A}$ (17) | na | 3.7$^{a,A}$ (0.4) |
| Forest | **Low aluminous clay–** | 0–5 | 493$^{ab,A}$ (99) | 68$^{ab,A}$ (19) | na | 210$^{a,A}$ (20) | 65$^{a,A}$ (22) | na | 115$^{b,B}$ (38) | 62$^{a,A}$ (25) | na | 150$^{c,B}$ (42) | 49$^{a,A}$ (25) | na | 33$^{b,B}$ (14) | 62$^{b,A}$ (36) | na | 4.8$^{a,A}$ (0.7) |
| | **High pedogenic Fe oxides** | 5–10 | 451$^{ab,A}$ (36) | 40$^{a,A}$ (11) | na | 210$^{a,A}$ (27) | 36$^{ab,B}$ (5) | na | 139$^{ab,B}$ (10) | 29$^{a,A}$ (7) | na | 166$^{b,B}$ (24) | 31$^{a,A}$ (11) | na | 34$^{a,B}$ (20) | 44$^{a,A}$ (18) | na | 4.6$^{a,A}$ (0.3) |
| Forest | **High aluminous clay–** | 0–5 | 604$^{a}$ (84) | 38$^{b}$ (5) | na | 140$^{b}$ (21) | 63$^{a}$ (34) | na | 100$^{b}$ (21) | 80$^{a}$ (51) | na | 125$^{c}$ (31) | 62$^{ab}$ (28) | na | 31$^{b}$ (13) | 101$^{ab}$ (59) | na | 4.3$^{ab}$ (0.4) |
| | **Low pedogenic Fe oxides** | 5–10 | 561$^{a}$ (47) | 26$^{a}$ (14) | na | 163$^{b}$ (12) | 28$^{b}$ (7) | na | 118$^{b}$ (17) | 22$^{a}$ (3) | na | 127$^{b}$ (21) | 25$^{a}$ (6) | na | 30$^{a}$ (1) | 43$^{a}$ (18) | na | 4.1$^{a}$ (0.2) |
| Forest | **High aluminous clay–** | 0–5 | 397$^{b,A}$ (91) | 86$^{a,A}$ (21) | na | 157$^{b,A}$ (27) | 89$^{a,A}$ (32) | na | 163$^{a,A}$ (32) | 99$^{a,A}$ (50) | na | 208$^{b,B}$ (36) | 91$^{a,A}$ (38) | na | 74$^{a,A}$ (14) | 133$^{a,A}$ (47) | na | 4.0$^{b,A}$ (0.6) |
| | **High pedogenic Fe oxides** | 5–10 | 474$^{ab,A}$ (139) | 35$^{a,A}$ (7) | na | 156$^{b,A}$ (27) | 33$^{ab,A}$ (4) | na | 146$^{ab,A}$ (41) | 30$^{a,A}$ (4) | na | 172$^{b,A}$ (61) | 34$^{a,A}$ (4) | na | 52$^{a,A}$ (26) | 51$^{a,A}$ (6) | na | 4.6$^{a,A}$ (1.0) |
| Cropland | **Low aluminous clay–** | 0–5 | 347$^{a,A}$ (69) | 20$^{b,B}$ (3) | -73 | 147$^{b,A}$ (13) | 21$^{c,B}$ (1) | -75 | 115$^{b,B}$ (4) | 17$^{c,B}$ (1) | -74 | 318$^{a,A}$ (52) | 11$^{c,B}$ (3) | -80 | 74$^{b,A}$ (12) | 24$^{c,B}$ (1) | -81 | 3.6$^{a,A}$ (0.5) |
| | **Low pedogenic Fe oxides** | 5–10 | 368$^{b,A}$ (28) | 20$^{b,B}$ (1) | -50 | 143$^{b,A}$ (8) | 22$^{b,B}$ (5) | -44 | 113$^{b,B}$ (10) | 17$^{b,A}$ (2) | -37 | 299$^{a,A}$ (15) | 11$^{b,B}$ (2) | -61 | 77$^{b,A}$ (1) | 24$^{c,A}$ (3) | -53 | 3.7$^{b,A}$ (0.2) |
| Cropland | **Low aluminous clay–** | 0–5 | 201$^{b,B}$ (39) | 47$^{a,A}$ (7) | -30 | 212$^{a,A}$ (12) | 49$^{a,A}$ (2) | -25 | 173$^{a,A}$ (18) | 42$^{a,A}$ (3) | -32 | 296$^{a,A}$ (33) | 46$^{a,A}$ (1) | -6 | 119$^{a,A}$ (4) | 62$^{a,A}$ (2) | ±0 | 2.7$^{b,B}$ (0.3) |



| | | | | | | | | | | | | | | | | | |
|---|---|---|---|---|---|---|---|---|---|---|---|---|---|---|---|---|---|
| | **High** pedogenic Fe oxides | 5–10 | 194[c,B] (11) | 47[a,A] (13) | +18 | 224[a,A] (15) | 49[a,A] (4) | +36 | 177[a,A] (1) | 42[a,A] (6) | +45 | 287[a,A] (13) | 45[a,A] (3) | +45 | 118[a,A] (29) | 58[a,A] (9) | +32 | 2.7[c,B] (0.1) |
| **Cropland** | **High** aluminous clay– | 0–5 | 296[ab,A] (40) | 26[b,B] (6) | -71 | 159[b,A] (8) | 29[b,B] (7) | -67 | 191[a,A] (2) | 28[b,B] (4) | -71 | 278[a,A] (25) | 35[b,A] (2) | -62 | 77[b,A] (10) | 41[b,B] (1) | -69 | 3.3[ab,A] (0.3) |
| | **High** pedogenic Fe oxides | 5–10 | 593[a,A] (95) | 25[b,A] (3) | -29 | 118[b,A] (21) | 26[b,B] (2) | -21 | 107[b,A] (29) | 25[b,A] (4) | -17 | 138[b,A] (37) | 32[b,A] (3) | -6 | 43[b,A] (10) | 41[b,B] (5) | -20 | 5.3[a,A] (0.6) |

na = not applicable.




In summary, mineralogical combinations and land use significantly affected the aggregate size

distribution of soils, despite quantitative relations to mineralogical proxies could not be observed for each
aggregate class. In undisturbed forest soils, higher pedogenic Fe contents resulted in increasing MWD
especially in 0–5 cm depth and significantly larger amounts of $> 2$ mm aggregates. The conversion from
forest to croplands either decreased MWD, as particularly observed for the low clay–high Fe combination,
or had no effect (low clay–low Fe). Overall, the observed differences in aggregate masses and MWD were
surprisingly moderate, given the widely differing contents in aluminous clay and Fe oxides across the
mineralogical combinations.

**3.3 Aggregate stability**
In general, there was little variation of MWD values for $> 4$ mm aggregates over all mineralogical
combinations. In fact, the MWD of this fraction was always close to its calculated mean diameter (6 mm;
calculation was done after *Youker* and *McGuinness* (1957)), overall indicating a high stability.
Nevertheless, there were some minor differences in aggregate stability across mineralogical combinations.
The low clay–low Fe and high clay–low Fe combinations had a significantly lower aggregate stability in
comparison with the two other combinations under the two land uses (Table 3). The slightly higher
abundance of 2–4 mm aggregates in the low clay–high Fe combination under forest and cropland was
accompanied by a significantly higher aggregate stability under both land uses (Table 2 and 3). In
summary, all aggregates can be classified as stable with only minor differences imposed by the
mineralogical combinations. Slightly higher aggregate stability was associated with a larger amount of
pedogenic Fe, and increasing $Fe_d$ to aluminous clay ratios, whereas differences in the amount of aluminous
clay had almost no effect on the aggregate stability (Table S2).






**Table 3:** Aggregate stability of selected aggregate size fractions after applying the fast wetting procedure along the different combinations of aluminous clay and pedogenic Fe oxides, indicated by the resulting mean weight diameter (MWD). Aluminous clay represents the weight sum of kaolinite and gibbsite present in the < 2-µm fraction after removal of OM and pedogenic Fe oxides. Lower case letters indicate significant differences within a certain land use separated by depth, and capital letters denote significant differences between land uses. Sample numbers for the combinations are as follows: ʻlow clay–low Feʼ under forest ($n = 4$), ʻlow clay–high Feʼ under forest ($n = 4$), ʻhigh clay–low Feʼ under forest ($n = 3$), ʻhigh clay–high Feʼ under forest ($n = 7$); all cropland combinations ($n = 3$).

| Land use | Mineralogical combination | Depth | MWD | |
|---|---|---|---|---|
| | | | Fast wetting > 4 mm | Fast wetting 2–4 mm |
| | | (cm) | (mm) | |
| **Forest** | **Low** aluminous clay– | 0–5 | **4.9**[b, A] (0.4) | **2.6**[b, A] (0.1) |
| | **Low** pedogenic Fe oxides | 5–10 | **5.1**[a, A] (0.3) | **2.4**[b, A] (0.3) |
| **Forest** | **Low** aluminous clay– | 0–5 | **5.6**[a, A] (0.2) | **2.8**[a, A] (0.1) |
| | **High** pedogenic Fe oxides | 5–10 | **4.9**[a, A] (0.9) | **2.7**[a, A] (0.1) |
| **Forest** | **High** aluminous clay– | 0–5 | **5.4**[ab] (0.4) | **2.7**[b] (0.0) |
| | **Low** pedogenic Fe oxides | 5–10 | **4.5**[a] (1.2) | **2.4**[b] (0.3) |
| **Forest** | **High** aluminous clay– | 0–5 | **5.5**[a, A] (0.2) | **2.6**[b, A] (0.1) |
| | **High** pedogenic Fe oxides | 5–10 | **5.2**[a, A] (0.4) | **2.6**[ab, B] (0.1) |
| **Cropland** | **Low** aluminous clay– | 0–5 | **4.4**[b, A] (0.1) | **2.6**[c, A] (0.0) |
| | **Low** pedogenic Fe oxides | 5–10 | **4.9**[b, A] (0.3) | **2.4**[b, A] (0.1) |
| **Cropland** | **Low** aluminous clay– | 0–5 | **5.2**[a, A] (0.2) | **2.9**[a, A] (0.0) |
| | **High** pedogenic Fe oxides | 5–10 | **5.3**[ab, A] (0.1) | **2.8**[a, A] (0.0) |
| **Cropland** | **High** aluminous clay– | 0–5 | **4.9**[a, B] (0.2) | **2.7**[b, A] (0.1) |
| | **High** pedogenic Fe oxides | 5–10 | **5.6**[a, A] (0.2) | **2.8**[a, A] (0.0) |

### 3.4 Organic carbon in soils and aggregate size fractions

Variation in mineral constituents caused different soil OC contents, ranging between 19 to 95 g OC kg$^{-1}$ across all sites including both land use and depth (Table 1). As outlined in *Kirsten* et al. (2021), significantly higher OC stocks were observed for low clay–high Fe combination under cropland and for



high clay–high Fe combinations under forest. Forest conversion to cropland caused marked OC losses for
the low clay–low Fe combination but no or minor losses for the low clay–high Fe combination (Table 1;
*Kirsten* et al., 2021).
A significant proportion of the total OC content of all forest soils was present in > 4 mm aggregates
in both depth increments (low clay–low Fe: 33% < high clay–high Fe: 43% < high clay–low Fe: 51% <
low clay–high Fe: 52%; Figure 1b). Forest to cropland conversion caused OC losses from most aggregate
size fractions (Figure 2). For the > 4 mm aggregates this was significantly modified by the mineralogical
combinations at least at 0–5 cm depth, generally following the order: low clay–high Fe < high clay–high
Fe < low clay–low Fe (Table S3). Losses of OC from aggregate size fractions were generally higher at 0–5
than at 5–10 cm depth (Figure 2). As mentioned above, no significant loss of total OC occurred for the
low clay–high Fe combination, irrespective of the significant decline of the > 4 mm aggregate fraction
(Table 2). Hence, OC formerly associated with large macroaggregates persisted the land-use conversion to
croplands residing in newly formed smaller aggregates. While there were differences in OC losses among
mineralogical combinations, there was little indication that coarser aggregate size fractions lost more OC
than smaller ones (Table 2).



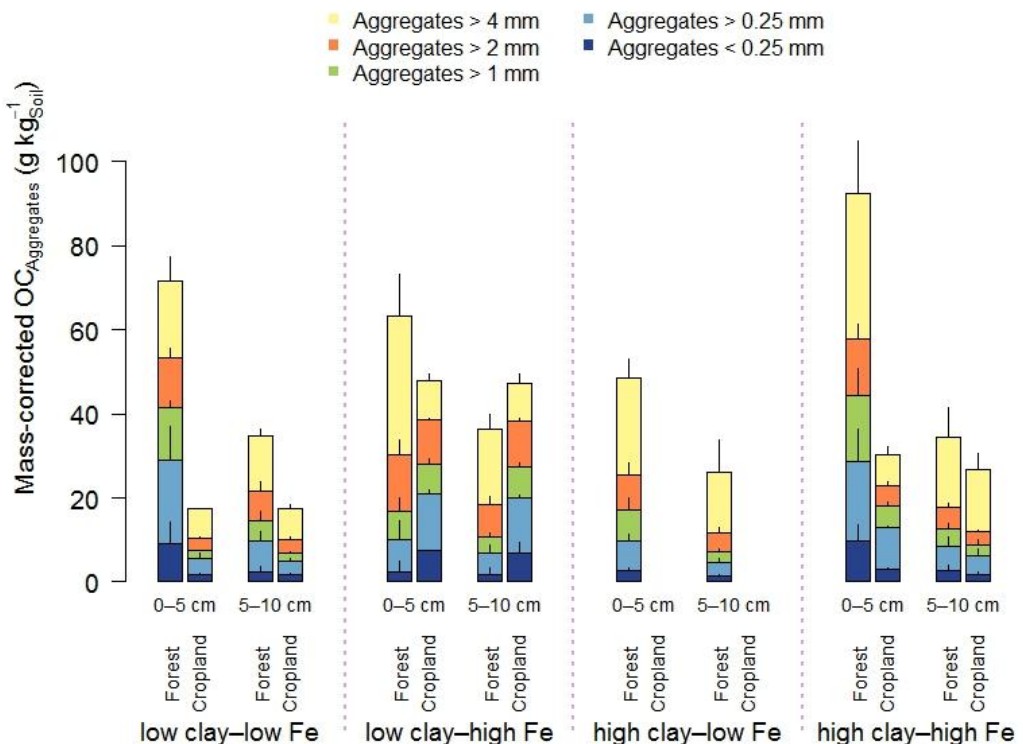

**Figure 2**: Mass-corrected OC contents of aggregate size fractions along the mineralogical combinations. Clay represents the weight sum of kaolinite and gibbsite present in the < 2-μm fraction after removal of OM and pedogenic Fe oxides, and Fe denotes the content of pedogenic Fe oxides extracted with dithionite-citrate-bicarbonate. Sample numbers for the combinations are as follows: 'low clay–low Fe' under forest ($n = 4$), 'low clay–high Fe' under forest ($n = 4$), 'high clay–low Fe' under forest ($n = 3$), 'high clay–high Fe' under forest ($n = 7$); all cropland combinations ($n = 3$).



## 4. Discussion


The aggregate size distribution of soils along the mineralogical combinations under both land uses were in
the range of values reported for African soils. For example, soils with strongly contrasting clay content
(220 and 650 g kg$^{-1}$) but similar clay mineralogy (kaolinite) in Kenya displayed macroaggregate contents
of 245 and 636 g kg$^{-1}$ soil, respectively (*Gentile* et al., 2010), and also high aggregate stability with MWD
values of the 2–4.6 mm aggregates ranging from 2.5 to 3.2 mm (*Kamamia* et al., 2021). These values are
close to those observed in our study soils for 2–4 mm aggregates. In contrast, soils in Brazil under native
forest vegetation and similar mineral composition (kaolinite, gibbsite, hematite) even subsumed over 90%
of total aggregate mass in > 2 mm aggregates (*Maltoni* et al., 2017). Nonetheless, reported data all point at
a better soil structure and aggregate stability of tropical soils dominated by low-activity clay minerals and
well-crystalline Fe oxides, which is consistent with all mineralogical combinations of this study.

### 4.1 Aggregation and aggregate stability as controlled by aluminous clay and pedogenic Fe oxides


Our data demonstrates that mineralogical combinations, with contents of aluminous clay varying by factor
three and pedogenic Fe oxides by factor five, did not result in entirely different aggregation and stability
patterns in the studied weathered tropical soils. Yet, we noticed some distinct modifications of the
aggregation size distribution and aggregate stability in both forest and cropland soils. The low clay–low
Fe soil under forest had a significantly smaller amount of > 4 mm and 2–4 mm aggregates and a
significantly lower MWD than all other mineralogical combinations. Notably, a combined increase in
aluminous clay and Fe oxides did not necessarily cause a shift towards larger aggregates and thus higher
MWD (see low clay–high Fe forest). Furthermore, the low clay–low Fe and high clay–high Fe
combinations under forest contained more < 0.25 mm aggregates. Thus, under undisturbed soil conditions
it appears that the formation of larger aggregates is promoted if one of the two aggregate-forming mineral
fractions is more abundant than the other (high clay–low Fe and low clay–high Fe combinations). The
high clay–low Fe and high clay–high Fe combinations under forest also nicely demonstrate how nearly
equal amounts of aluminous clay plus pedogenic Fe oxides (i.e. similar clay contents) cause different



amounts of > 4 mm aggregates. Consequently, the connection between textural properties and aggregation
can remain hidden (*Barthès* et al., 2008) without considering the mineralogical composition of the whole
clay fraction (*Fernández-Ugalde* et al., 2013; *King* et al., 2019; *West* et al., 2004).

Land-use change had a distinct impact on aggregate distribution like indicated in other studies

(*Feller* and *Beare*, 1997; *Six* et al., 2002) and depended also on the mineralogical combinations, though
croplands not followed the trajectory observed under forest. A significantly lower MWD under low
clay–high Fe rather than low clay–low Fe can be mainly attributed to a reduced amount of > 4 mm
aggregates. We assume that differences in the ratio of pedogenic Fe to aluminous clay in the low clay–low
Fe and high clay–high Fe (0.13 to 0.15) in comparison with the low clay–high Fe combination (0.47 to
0.51) under cropland explains the stability of 'card-house' structures like described for mineralogically
similar Oxisols from Brazil and India (*Bartoli* et al., 1992). Accordingly, a higher $Fe_d$ to aluminous clay
ratios seems to be disadvantageous for the formation of such structures, especially in > 4 mm aggregates.
The different pH-dependent charge characteristics of kaolinite and pedogenic Fe oxides (*Kaiser* and
*Guggenberger*, 2003), and their relative share can lead to altered charge properties of soils (*Anda* et al.,
2008). We hypothesize, that an increasing amount of Fe oxides in the investigated mineralogical
combinations adds more positive charge, thus possibly reducing structural integrity and aggregate stability
if not sufficiently compensated by OM or clay minerals. Furthermore, in the low clay–high Fe cropland
combination, land-use change caused a significant four-fold increase of < 0.25 mm aggregates due to the
breakdown of > 4 mm aggregates. Nonetheless, our results show that agricultural management does not
necessarily decreases macroaggregation and related MWD's, like reported elsewhere (*Rabbi* et al., 2015).

The dominant role of pedogenic Fe oxides for macroaggregation under undisturbed tropical soil

conditions proposed by *Six* et al. (2002) cannot be confirmed in our study. This is because the low
clay–high Fe forest soil contained a smaller amount of > 4 mm aggregates compared to the high clay–low
Fe forest soil in both depth increments. Consequently, this rather points at the importance of kaolinite for
macroaggregation, which is in line with results from two Oxisols in Brazil (*Vrdoljak* and *Sposito*, 2002),
showing kaolinite being the backbone of the investigated aggregate size fractions. The less intense





formation of > 4 mm aggregates in the low clay–high Fe forest combination was also observed under
cropland, whereas the low clay–low Fe and high clay–high Fe croplands showed either no significant
decrease or even an increase in > 4 mm aggregate mass. Thus, simultaneous abundance of large amounts
of aluminous clay and pedogenic Fe oxides preserved a higher aggregate stability than under
mineralogically imbalanced conditions, although no conclusions can be drawn for the high clay–low Fe
combination. Nonetheless, > 4 mm aggregates had a higher resistance to field operations in mineralogical
combinations with lower $Fe_d$ to aluminous clay ratios (0.13 to 0.15).

In contrast to the > 4 mm aggregates, 2–4 mm aggregates corresponded more clearly to the positive

effect of pedogenic Fe oxides on aggregation and aggregate stability as proposed for weathered tropical
soils (*Igwe* et al., 2013; *Peng* et al., 2015; *Six* et al., 2002). Both, the low clay–high Fe forest and low
clay–high Fe cropland soils contained somewhat but significantly more 2–4 mm aggregates than other
mineral combinations in concert with a higher aggregate stability of this particular fraction. This finding
also demonstrates that mineral interactions forming water-stable aggregates in tropical soils are differently
affected by a given mineralogical combination. Higher $Fe_d$ to aluminous clay ratios (> 0.45) modulate
aggregate distribution towards aggregates 2–4 mm, whereas distinctly lower values (high clay–low Fe
forest: 0.12) shifted the maximum to > 4 mm aggregates. Overall, the two macroaggregate fractions
discussed above are differentially affected by the mineralogical combinations, although the magnitude was
less than expected, given the pronounced variation in aluminous clay and Fe contents.

**4.2 Importance of aggregation for OC persistence – effects of aluminous clay and pedogenic Fe ox-**
**ides**
Clay minerals and Fe oxides are considered as important mineral constituents fostering aggregation and
subsequent OC storage via physical protection (*Denef* et al., 2004). The overwhelming portion of OC in
the studied topsoils resided in mineral-organic associations (35−81%), whereas OC occluded in
aggregates amounted to 7−24%, with a lower share under cropland than forest as determined by density
fractionation (*Kirsten* et al., 2021). The low clay–high Fe cropland had an OC content more than twice



larger than that of the low clay–low Fe cropland, but comprised a significantly smaller MWD. Thus, a
shift towards more macroaggregation, indicated by a larger MWD in certain mineralogical combinations,
did not result in higher total OC storage, like shown for other tropical soils (*Barthès* et al., 2008; *Bartoli* et
al., 1991; *Spaccini* et al., 2001). The OC content of the > 4 mm aggregate and 2–4 mm aggregate fractions
accounted for 42 to 73% of the total soil OC content (Figure 1b). This, however, does not *per se* indicate
the relevance of macroaggregation for OC storage in weathered tropical soils like proposed by others
(*Feller* and *Beare*, 1997; *King* et al., 2019; *Six* et al., 2002). The high clay–low Fe forest with the highest
share in > 4 mm and 2–4 mm aggregates had significant lower OC contents in these fractions than most
other mineralogical combinations. Furthermore, if land-use change is taken into account, we observed
significantly reduced OC contents in the majority of macroaggregate fractions of the low clay–low Fe and
high clay–high Fe croplands, as reported in other studies (*Blanco-Canqui* and *Lal*, 2004; *Lobe* et al.,
2011). In contrast, least changes of aggregate-associated and total soil OC contents was observed in the
low clay–high Fe combination, despite it experienced the strongest disaggregation of the largest
macroaggregates (Figure 1a and Figure 2). We conclude that larger amounts of > 2 mm aggregates or
higher stability during wet sieving not automatically translates into higher aggregate-associated OC
contents, as reported for Ferralsols (*Maltoni* et al., 2017). Given all these observations and the fact that
occluded OM determined by density fractionation was mostly of subordinate relevance, particularly in
croplands, OC storage in study soils seems rather disconnected from their aggregation status.
Consequently, the loss of large aggregates and the mass redistribution into smaller aggregate size fractions
does not automatically imply a loss of soil OC, because a substantial part of the OC in aggregate fractions
is bound to minerals with a higher persistence against land-use change (*Kirsten* et al., 2021). Here, density
fractionation could shed more light on the nature and quantity of OM located in certain aggregate size
fractions.

Microaggregates contained the highest OC content per unit of mass for almost all mineralogical

combinations, depth increments, and land uses (Table 2). This is in line with the findings of *Chenu* and
*Plante* (2006) and *Lobe* et al. (2011) that microaggregates can significantly contribute to OC storage. As



aggregates were isolated by dry sieving, these microaggregates were not located inside larger aggregates,
rendering them principally better accessible for OC allocation. Particularly OC contained in the
< 0.25 mm aggregates of the low clay–high Fe combination revealed a strong persistence against land-use
change, which explains well the unaltered soil OC contents upon land-use change.





## 5. Conclusions

Classification of soils into mineralogical combinations of aluminous clay and pedogenic Fe oxides revealed significant effects of mineral constituents on soil structure and related OC storage in weathered tropical soils. Despite that, overall patterns across combinations were more similar than different, *i.e.*, always comprising a high level of macroaggregation and aggregate stability. Aggregates > 4 mm of the low clay–low Fe and high clay–high Fe combinations were less affected by land-use change, thus pedogenic Fe in a certain relation with aluminous clay (0.13 to 0.23) seems beneficial to maintain the structural integrity of macroaggregates. Despite the high physical stability, OC contents of macroaggregates declined substantially in most mineralogical combinations during forest–cropland conversion. This highlights the fact that structural integrity of macroaggregates during land-use change cannot be equated with OC persistence. For the low clay–high Fe combination, substantial destruction of > 4 mm aggregates during land-use change due to agricultural management was also not accompanied by higher OC losses. Thus, we have to reject our initial assumption that the mineralogical combination resulting in the largest aggregate stability better preserved OC during conversion of forests into croplands. We suggest that in weathered tropical soils this is largely attributable to the importance of mineral-organic associations, where changes in aggregation do not immediately offset the stabilizing effect of soil minerals.

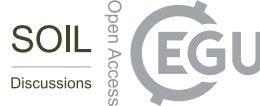

## 7. Author contribution

KK, RM, MK, and KHF designed the project. MK, KK, RM, DNK, and KHF collected soil or data to supported the sampling campaign. MK, KK, RM, and KHF evaluated data and all authors conducted a thorough critical review of the manuscript. MK, KK, and RM wrote the manuscript with contribution of all authors.

## 8. Competing interests

The authors declare that they have no conflict of interest.

## 9. Acknowledgements

We are grateful to the officials of Amani Nature Reserve who supported the field campaign in February 2018. Aloyce Mkongewa enthusiastically assisted fieldwork. We are also indepted to Gisela Ciesielski, Manuela Unger, Mandy Meise, Tobias Krause, Thomas Klinger, Gudrun Nemson-von Koch, and Christine Krenkewitz for laboratory support and analytical work. This study was supported by grants of the Deutsche Forschungsgemeinschaft (DFG): FE 504/15-1, KA 1737/16-1, and MI 1377/11-1.



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
