# Peer review of "Aluminous clay and pedogenic Fe oxides modulate aggregation and"

_SOIL, 2020_

## Author Comment (AC2)

Dresden University of Technology – Dept. Soil Science and Site Ecology Pienner Straße 19 • 01737 Tharandt, Germany

To the Editorial Board Copernicus Publications Journal Soil

Dear Editorial Team,

Thank you very much for accepting our manuscript for open discussion. We found the comments and questions by the unknown referee #2 very useful and suggest revisions of the manuscript accordingly. All of our responses are listed below.

**Response letter**

Response to the comments and questions of the referee's related to the manuscript:

**MS No.: soil-2020-98**

Special Issue: Tropical biogeochemistry of soils in the Congo Basin and the African Great Lakes region

With the title:

"Aluminous clay and pedogenic Fe oxides modulate aggregation and related carbon contents in soils of the humid tropics"

**Response to the first general comments of referee 1.**

**Referee 1:** "In my first very general comment, it seems that there is some overlap with the study published by the same first author Kirsten et al; 2021 "Iron oxides and aluminous clays selectively control soil carbon storage and stability in the humid tropics" Scientific reports, 11, 5076."

**Our response:** The overlap between both studies is intentional, because it based on the used approach to indentify the influence of aluminous clay (kaolinite, gibbsite) and pedogenic Fe oxides (goethite and hematite) on aggregate distribution and associated organic carbon (OC) storage and persistence. We used the same field sites (i.e. mineralogical combinations) as in Kirsten et al. (2021) who studied the influence of mineral-organic associations on OC storage and persistence. In contrast to this previous paper, we evaluated the effects of aluminous clay and pedogenic Fe oxides as the two most important constituents of the clay-sized fraction in weathered soils on aggregation and the consequences for OC persistence after land-use change.

**Referee 1:** "*I recommend the authors to clarify how the results of aggregates are new and add novelty compared to Kirsten et al. 2021.*"

**Our response:** In our manuscript, we went far beyond studying the effects of clay content on aggregation. We disentangled the individual role of aluminous clay and pedogenic Fe oxides for determining aggregate size distribution and aggregate stability and the consequences for OC persistence after land-use change. In contrast, Kirsten et al. (2021) focused on mineral-organic associations and how they influence OC storage and persistence.

**Referee 1:** "I mean here it makes sense to clarify this point as much as possible as aggregate size fraction and OC distribution in these fractions is one of the main control on soil carbon storage and stability in soils. So it makes me thinking that the data of Kirsten et al. 2021 must be presented, treated and interpreted together with the present data about aggregation."

**Our response:** As both studies used the same field sites (i.e. mineralogical combinations), we refer as much as possible to the study of Kirsten et al. (2021) in order to avoid any unnecessary overlap whereas all necessary data were presented and used in the discussion. In the response to the next comment of referee 1 details are given where and what data have been used already published by Kirsten et al. (2021).

**Referee 1:** "In my opinion, it could be great to build your research question based on what you uncover in Kirsten 2021, because their results can be a solid foundation to this study. So summarizing and building on Kirsten 2021 in the introduction could serve to expose the novelty of the present study."

Our response: We partially used the data and findings made by Kirsten et al. (2021) and will add new sentences to the text: In the introduction (Lines 129-131), material and methods (Lines 189–191; 224), and discussion (Lines 547–551; 573–575) for preparing the fundament of the study and contribute to our discussion. We further tried to use other studies conducted in the humid tropics to highlight the partially contradicting results about the mechanistic understanding of the role of clay content (mineralogy) on aggregation and its contribution to OC storage and persistence under contrasting land uses. For further clarification of our research question, we will add the following sentences (Lines 149-152: "In the precursor study, we found a positive relationship between the storage of mineral-associated OC and the ratio of pedogenic Fe to aluminous clay under forest and cropland land use, suggesting that a larger share of Fe oxides is linked to larger OC storage and persistency against land-use change (Kirsten et al., 2021). In the present study, we test whether aggregation and its contribution to OC storage follow similar patterns, or are decoupled from the individual contribution main mineral constituents. In detail, our main research goal was to investigate the individual role of aluminous clay and pedogenic Fe oxides in contrasting combinations determining...").

**Referee 1:** "I also recommend to clarify the results interpretation without comparing the two ecosystems, especially because the co-variable inducing OM changes due to land use and management practices. I am saying the bottom line of the study is to compare the two ecosystems, but the current presentation of results and data interpretation make it a bit fuzzy, confusing."

**Our response:** The high variability in the effects of land-use changes on soil OC (e.g. Don et al., 2011) motivated us to study the controls of OC persistence to land-use changes. In the present manuscript, we aimed to improve knowledge about how the storage and persistence of OC in defined aggregate size fractions is affected by different combinations of aluminous clay and pedogenic Fe oxides and what the consequences are for OC persistence after land-use change from natural forest to cropland. Therefore, the comparison between the two land uses is an essential part of the entire study. However, we presented and discussed the results first for natural forests and second for the studied cropland sites, i.e. addressing effects of land-use

change. That will be further improved (whenever possible) during the revision. With this approach, we are able to test whether aggregation and the related OC in aggregate size fractions stays constant under both land uses in the respective mineralogical combination – an important indication of OC persistence. For example, when the mean weight diameter and OC storage in aggregate size fractions within a given mineralogical combination remains similar under both forest and cropland, this would indicate that this specific mineralogical condition favors aggregate stability and OC persistence. On the other hand, once the mean weight diameter and OC storage in aggregate size fractions decline in a given mineralogical combination upon land-use change, this would rather suggest that aggregate integrity and OC persistence is less supported by the mineralogical condition.

**Referee 1:** "I know the sites and sampling, and some of the methodologies are already presented in Kirsten et al. 2021, but given the topic of the submitted study I strongly recommend the authors to make the site selection and soil sampling crystal clear to help the readers to understand how environmental factors are similar between the studied sites under forest (how is your vegetation homogenous) and croplands (especially here for agricultural practices)."

**Our response:** We agree with referee 1 that providing more information about site selection would be beneficial for the reader. We will add additional information to the material and method section to (i) describe the selection of the study sites, (ii) illustrate that the forests have similar input of fresh organic matter input, and (iii) show the similar management practices at the different croplands (Lines 190–197: "The site selection was done based on total clay content determined in the field and the associated total Fe content measured with a portable XRF device (Kirsten et al., 2021). We did not observe systematic differences in vegetation composition of the forest sites and NMR spectra showed a similar composition of litter for each of the two land uses investigated (Kirsten et al., 2021). Furthermore, several visits in the study region over the last decade (2012, 2013, 2015, and 2018) combined with personal communications with farmers and local partners working in the region, enabled us to select cropland sites with similar agricultural management (cultivation of cassava (*Manihot esculenta*), hand hoe tillage, biomass burning before seed bed preparation).").

**Answers to detailed comments of referee 1**

**Abstract:**

**Referee 1:** "*Line 27*: could you please clarify what you mean by "positive feedbacks on soil carbon storage"."

**Our response:** We intended to emphasize the positive effect of aggregation on soil OC storage. We will change "feedbacks" into "effects" (Line 27).

**Referee 1:** "Line 30: would it make sense to use either "aluminous clays" or "aluminosilicates" for the sake of clarity? I would prefer "aluminosilicates"."

**Our response:** Beside the dominant aluminosilicate kaolinite  $(Al_2Si_2O_5(OH)_4)$ , we also identified the hydroxide gibbsite  $(\gamma$ -Al(OH)3) in the clay fractions of the study soils (Kirsten et al., 2021). This is the reason why we summarized both mineral constituents under the term "aluminous clay" and would like to keep with this denomination. We will also correct a mistake in Line 30 ("as both secondary aluminosilicates and Fe oxides…" into "as aluminosilicates, aluminum hydroxides, and Fe oxides…"

**Referee 1:** "(*Line 37* should be clarify, a bit wordy)  $\rightarrow$  and Lines 38–41: I recommend to reword this sentence as this is not clear why you oppose formation of large macroaggregates and promoted OC storage and persistence."

**Our response:** We will clarify our statement in Lines 37–38: "Patterns in soil aggregation were rather similar across the different mineralogical combinations (high level of macroaggregation and high aggregate stability)." Furthermore, we will add in Lines 38–39: "Nevertheless, we found some statistically significant effects of aluminous clay and pedogenic Fe oxides on aggregation and OC storage." The proposed reformulation and expansion of the abstract will lead more clearly to the following sentences, in which we explain why we assume that the mineral properties control the aggregation and storage of OC, but that the two processes (aggregation and OC storage) were not closely related to each other. Therefore, we would like to keep these important findings in the abstract and also clearly announce the threshold values to distinguish between the selected mineralogical combinations.

**Referee 1:** "Line 42: "low clay-high Fe" does not ease the reading. I would suggest to present it another way to read smoother"

**Our response:** We agree to the comment of referee 1, in order to improve readability. Therefore, we will not use these acronyms in the abstract. Furthermore, we will rephrase the sentence (Lines 43–45: "The combination with low aluminous clay and high pedogenic Fe contents displayed the highest OC persistence, despite conversion of forest to cropland caused substantial disaggregation.")

**Referee 1:** "Line 36: a bit awkward as mineral-organic interactions are part of the aggregation. How can you oppose them?"

**Our response:** We assume that the referee means Lines 45–48. We agree with the statement that mineral-organic associations are part of aggregates (building blocks) as shown e.g. by Totsche et al. (2018). Although mineral-organic associations are part of the various aggregate fractions, an additional part of (particulate) OC is also occluded within the aggregates potentially contributing to OC storage, which could be considered as an indirect effect of mineral-organic associations. With our last sentence, we want to emphasize that aggregation is less important for OC persistence in comparison to the more direct effects of mineral-organic associations. Therefore, we would like to stay with this sentence.

**Introduction:**

**Referee 1: "Line 61: I suggest to change "reacting" by "associating"**

**Our response:** We would be happy to keep "reacting" because that word should express the more active role of the charges for these interactions to form an association.

**Referee 1:** "Line 64–65: reading this sentence makes me thinking – how does it make sense to think aggregation processes in soils could be associated with one unique mineral phase? As long as soil is multiphase, it seems pretty reasonable to assume aggregation is explained by interactions between various phases. While I fully understand the need to better understand how proportion between minerals play a role in aggregation. Maybe, the sentence here needs to be rephrased."

**Our response:** We agree with the statement of referee 1. We will rephrase the sentences as follows (Lines 84–87): "Aggregation might be ascribed to inorganic or organic cementing agents with no consensus about the relevance of each individual agent. Understanding the effects of individual cementing agents for aggregation is needed to disentangle their potential

contribution to soil aggregation.". Additionally, we already described various influencing factors on aggregation (Lines 87–111).

**Referee 1:** "Lines 83–86: In addition could you please clarify how you isolate the OM content and quality between your sites? This is neat the idea to choose sites with identical mineralogical context. But OM quantity and quality play also a role in aggregation, so that it could make sense to explain whether this variable is also similar between studied sites."

**Our response:** We analyzed the OC contents of each aggregate size fraction and bulk soil by high temperature combustion at 950°C and thermo-conductivity detection (Vario EL III/Elementar, Heraeus, Langenselbold, Germany). Furthermore, we applied 13C-NMR spectroscopy to test the composition of OM along the mineralogical combinations (Kirsten et al., 2021). We considered the quantity of OC mainly as consequence of the various mineralogical combinations as well as land-use, as OC associated with minerals often was the most prevalent carbon form in the study soils (Kirsten et al., 2021). Furthermore, reactive minerals are well-known "filters" that can preferentially retain high-affinity OM components while others, less sorptive ones are excluded from sorption (sorptive fractionation). Consequently, the contents and composition of OM in mineral-organic associations, especially in topsoils as in this study, can be very similar despite of pronounced differences in aboveground litter quality (Mikutta et al., 2019). The composition of litter varied to a certain extent as indicated by the NMR spectra but, as mentioned before, we did not consider these differences in litter composition as decisive control of OM accumulation and stabilization (cf. Schmidt et al., 2011; Lehmann and Kleber, 2015).

**Referee 1:** "*Line 92*: need to precise here what you mean by "to which extent aluminous clay and pedogenic Fe oxides" do you mean, the proportion? The type of oxides and aluminosilicates?"

**Our response:** We refer to the amount (g kg-1 soil) of aluminous clay (kaolinite + gibbsite) or pedogenic Fe oxides (goethite + hematite) as proxies for the two mineralogical categories. To enhance clarity, we will rephrase the sentence (Lines: 125-127: "We are currently not aware of any studies that solve the puzzle of the extent to which the amount of aluminous clay and pedogenic Fe oxides controls soil aggregation and OC storage in highly weathered soils of the humid tropics.").

**Material and methods:**

**Referee 1:** "Line 141: I am aware fractionation methods are time consuming but could the authors explain why they do not investigate aggregate size under 250µm and also why the authors do not measure oxalate-extractable and DCB-extractable Fe and Al in each fractions, so that to be able to have direct relationships between type of mineral in aggregate fraction and its contribution to the OC pool."

**Our response:** With our study we aim at focusing on larger aggregate fractions because landuse change has a particularly strong impact on macroaggregates (John et al., 2005; Lobe et al., 2011; Maltoni et al., 2017). Thus, if the mineralogical composition has a distinct impact, it should be most pronounced for macroaggregates. We are not aware of any study that investigated into the distribution of individual mineral phases in aggregates > 0.25 mm compared to the bulk soil. As this would have been an extremely laborious task, we saved the resources to measure the dithionite-citrate-bicarbonate extractable-Fe and the aluminous clay for each individual aggregate fraction. However, this might provide valuable information in future studies.

**Referee 1:** "Regarding the specific extraction, it is not clear how the authors can relate aluminous minerals to oxalate-extractable Al which is very specific to short-range ordered minerals. Authors refer to Kirsten et al. 2021 for the method to determine aluminous clay based on DCB extraction and textural analysis. It could help to summarize here how they proceed, especially because all interpretations in the paper depend on this quantification."

**Our response:** We agree with referee 1 that ammonium-oxalate extractable Al is representative for short-ranged ordered Al minerals. We set this into relation with our total Al element contents determined by XRF (Kirsten et al., 2021) in order to demonstrate that nearly all Al-bearing mineral phases are crystalline ones and relate to residual primary minerals from the parent material, kaolinite and gibbsite. To clarify on that, we will provide the reader with a thorough understanding of the methodology used to distinguish between the four selected mineralogical combinations (Lines 218–224: "Briefly, 5–6 g soil pre-treated with 30% H2O2 were extracted with 30 g sodium dithionite (Na2S2O4) and 1.35 L buffer solution (0.27 M trisodium citrate dihydrate (C6H5Na3O7 • 2H2O) + 0.11 M sodium bicarbonate (NaHCO3)) at 75°C in a water bath for 15 min (Mehra and Jackson, 1958). The Fe concentration of the extracts were measured by inductively coupled plasma optical emission spectroscopy (ICP-OES) using a CIROS-CCD instrument (Spectro, Kleve, Germany). The residues of the

extraction were then subjected to a texture analysis using the pipette method (Gee and Bauder, 1986).").

**Referee 1:** "I recommend this paper to be self-sufficient concerning the description of site location. As authors are dealing with aggregate processes, it is crucial to ensure all soil characteristics are strictly identical sites, except of course for the gradient in Fe and Al phases content. It could be helpful also to get some words explaining a bit what is behind the scene with regard to the mineralogical changes. What is the soil-forming processes and factors responsible for these changes?"

**Our response:** As already proposed we will add more information on the study site (including agricultural management) and the applied analyses in addition to the reference of Kirsten et al. (2021). We can only speculate about the reasons for the observed differences in the amounts of aluminous clay and pedogenic Fe oxides. Possible explanations include slight spatial variations in bedrock materials (mafic biotite-hornblende-garnet gneiss in the whole area) and/or variable weathering/desilification rates and/or variable extents of soil erosion and/or different stages of clay translocation. Desilification can lead to a relative enrichment of pedogenic Fe oxides compared to aluminous minerals such as kaolinite. The "good" soil structure along the mineralogical combinations promotes downward translocation of clay/dissolved ions with soil water into deeper soil horizons. We found clear indication for clay illuviation into deeper soil horizons (cf. Kirsten et al., 2019). This process could also be selective for certain mineral fractions, because of the pH-dependent charge characteristics of dominating clay-sized mineral phases (Kleber et al., 2015). Each of these processes might have contributed to observed variations in aluminous clay and F oxide contents.

**Results:**

**Referee 1:** "*Line 250*: which one takes over – mineralogical combination or land use?"**

**Our response:** Our data clearly show that both factors have a combined impact on aggregate distribution. Nonetheless, mineralogical composition is the underlying determinant controlling aggregate distribution. As stated in Lines 373–375: "The conversion from forest to cropland either decreased MWD, as particularly observed for the low clay–high Fe combination, or had no effect (low clay–low Fe)."

**Referee 1:** "Lines 269–271: ok it makes sense, I am just wondering how agricultural practices can affect aggregate stability compared to less managed forest ecosystem. This is pretty well documented and in your study I am wondering if Fe and Al phases can take over land use management when studying parameters such as aggregate stability. I am thus wondering if it makes sense to compare the two ecosystems. What do you think about interpreting the controls of mineral phases on aggregated inside each ecosystems without venturing into comparison between ecosystems."

**Our response:** Please, refer to our previous comment (response to comment 5). As described above, we have tried to always choose the same order of presentation of results and discussion. We found distinct effects of certain mineralogical combinations (e.g. low aluminous clay and high pedogenic Fe) on aggregate distribution independent from land use as well as significant but very small differences for aggregate stability between land uses, i.e. land use effects. Consequently, we want to keep both aspects for interpretation of our results.

**Referee 1:** "Line 286: how can you directly associate a variation in soil OC content to mineral constituents as land use and management practices can significantly affect OC. Again I would separately present the results for the two ecosystems, forest and cropland."

**Our response:** As explained in previous comments, our study was conducted in an area with very similar natural conditions except the contents the soils' contents in aluminous clay and pedogenic oxides. Our approach is based on the reasonable assumption of similar agricultural managements at all study sites, rendering agricultural management no decisive factor to explain variations in soil OC among arable soils. Nevertheless, OC contents were quite variable in the forest soils, potentially reflecting local variations in vegetation and soil moisture. Despite of this variability, aluminous clay and pedogenic Fe oxides explained a large variability of soil OC (Kirsten et al., 2021), confirming observations in the literature (Barthès et al., 2008; Coward et al., 2017; Lawrence et al., 2015). We totally agree that land-use change has an effect on OC and we related the extent of this effect to differences in aggregation for each mineralogical setting (defined by aluminous clay and Fe oxide contents). Nevertheless, we show important results for each individual land use and in direct comparison, so that the reader gets a comprehensive overview.

**Referee 1:** "*Line 295*: linked to my previous comment (*Line 286*) it is pretty confusing to read that ">4mm aggregates this was significantly modified by the mineralogical combination"

while OC input and quality (together with the way OM is processed in these two highly contrasting ecosystems) can also play a key role."

**Our response:** Please see the replies to the comments above. Here, we refer to the OC loss within this particular aggregate size fraction by conversion of forests into croplands, which depends, directly or indirectly, on aluminous clay and pedogenic Fe oxide contents.

**Discussion and Conclusions:**

**Referee 1: "Line 326: what do you mean by "did not result in entirely different""**

**Our response:** With this introductory sentence, we would like to emphasize that all soils were well aggregated and had a high aggregate stability. The observed statistically significant differences were rather small despite a large variation in the combination of aluminous clay and pedogenic Fe oxides. We will modify this sentence accordingly (Lines: 460–462). "Our data demonstrates relatively small differences in aggregate among the generally well-aggregated study soils, being characterized by high aggregate stability despite of large variations in aluminous clay (factor three) and pedogenic Fe (factor five) contents."

**Referee 1:** "*Line 333–335*: ok, it is the observation you did concerning your results, but how can you explain soils need a mineral phase take over the other one to promote aggregation. I am curious to learn a bit more here, maybe with the help of the state-of-the-art knowledge already published in this research field?"

**Our response:** We agree with referee 1, but we can only speculate based on published literature. We will add additional information to the main text (Lines: 482–493: "We assume that the positive effect of increasing aluminous clay content on the aggregate mass > 4 mm is related to the hybrid electrostatic properties of kaolinite on edges (variable) and surfaces (permanent negative), which enable the formation of characteristic cards-house structures (Qafoku and Sumner, 2002). In addition to this increase in aggregation caused by the dominance in kaolinitic properties (i.e. high clay–low Fe), we also expect that, similar to the study by Dultz et al. (2019), there are mixing ratios between aluminous clay and pedogenic Fe minerals, which lead to improved aggregation (greater MWD; i.e. low clay–high Fe). This effect is probably explained by changes in the electrostatic properties of the mineralogical combinations, as was shown in the study by Hou et al. (2007) for kaolinite in different relative combinations with goethite and hematite. Nevertheless, aluminous clay is the decisive control for macroaggregation in these weathered tropical soils, confirming the often described

promoting effect of increasing clay content on aggregation (Feller and Beare, 1997). Furthermore,...").

**Referee 1:** "Line 341: I am definitely uncomfortable with the study of land use effect on aggregate distribution through the lens of mineralogical variations. I think on top of mineralogical differences, the land use and management practice explain all the differences with regard to aggregate and OC distribution between forest and cropland."

**Our response:** As we discussed above, land use and management certainly have an impact on soil OC and the level of aggregation. But the magnitude of this effect – and this is what we emphasize – is modulated by the quantitative abundance of mineral constituents. Therefore, we think there is no real discrepancy in our views.

**Referee 1:** "Line 348: to be able to say that higher Fed/Al ratio control aggregate formation, I think you have to ensure there is no other effect concerning agricultural practices. I mean here: how are you sure that tillage, crop rotation, cover crops... are identical between your studied croplands?"

**Our response:** As mentioned above, we paid the utmost attention to sample cropland sites with similar tillage, cultivated crops, and a similar procedure regarding the management of crop residues. That does not mean that all of the single management operations were identical but at least they are similar – the best approximation which is feasible under field conditions.

**Referee 1:** "Line 385–386: it is part of the introduction, and what you are presenting from Kirsten could help to introduce your research question by presenting it in the introduction, in order to streamline the presentation of your objectives, and their novelties compared to Kirsten 2021."

**Our response:** We would prefer to keep this introduction into the new section of the discussion, which in our opinion eases the understanding for the readership. As stated above, we will expand the introduction by our previously published results (Kirsten et al., 2021) to better justify the objectives of this study.

**Referee 1:** "*Line 398*: taking into account land use changes as an explaining variable, compared to mineralogical changes is a bit "adventurous"."

**Our response:** With our data we show that land-use change has a distinct impact on OC storage, but the extent is significantly modulated by the selected mineralogical combinations.

**Referee 1:** "Lines 433–434: I agree but I think it could make better sense to only study the effect of mineralogical changes for each ecosystems, separately. It will help the reader to better catch your message regarding the role of Fe/Al ratios on aggregation formation for either forest ecosystem or croplands."

**Our response:** As outlined above, we think that the differences between forest and cropland sites in terms of the effect of aluminous clay and pedogenic Fe oxides on aggregation and associated OC is a key point of this study.

**Referee 1:** "*Lines 435–437*: I am afraid I do not understand your last conclusion sentence. *Need to be rephrased, IMO.*"**

**Our response:** We agree with referee 1 and will rephrase the sentence (Lines 598–603: "Hence, we must reject our initial hypothesis that the mineralogical combination that results in the greatest aggregate stability best preserves OC during the conversion from forest to cropland. Thus, the formation of macroaggregates cannot be considered as a main stabilization process for OC in strongly weathered soils of the humid tropics. We suggest that the formation of mineral-organic associations as part of the aggregate size fractions is the most important process that preserves OC during land-use change in these soils.").

**Response to the first general comments of referee 2.**

**Referee 2:** "In general, I think that the story of the manuscript, the hypotheses, and objectives need to be further improved. The use of highly specific jargon in the text makes challenging to understand the manuscript without enough context for some of the technical terms used. Being the scope of this journal so broad, I suggest reducing the jargon, use the same terminology across the manuscript and explain it briefly where possible. Finally, I agree with reviewer1 that this manuscript needs to include enough information to be a standalone manuscript and needs to explicitly state how it builds up from the previous study."

**Our response:** We appreciate the arguments of Referee 2 and will avoid very specific technical jargon and acronyms as much as possible. However, we rely on the use of specific terms as e.g. "aluminous clay" in order to be unambiguous in our language. We checked the correct use of technical terms and we will write out the abbreviations of the mineralogical combinations as much as possible to improve readability (e.g. Abstract: Lines 43–44). To further improve understanding of the used abbreviations, we will also add the definition of thresholds to differentiate between "high" and "low" levels of aluminous clay and pedogenic Fe oxides (Lines 226–228: "The threshold values for aluminous clay and pedogenic Fe oxides in order to distinguish between "high" and "low" were set to 250 g kg-1 and 60 g kg-1, respectively.").

**Referee 2:** "The Introduction does a good job exposing the gaps in knowledge and presents enough information to support the proposed hypotheses. However, it lacks context regarding some of the factors assessed. For example, why is it relevant to measure the response variables at different depths (0–5/5–10cm)?"

**Our response:** Soil organic carbon storage in the study region is most influenced by cultivation and other environmental factors in the uppermost soil depths (0-10 cm) (Kirsten et al., 2019). Furthermore, change in soil structure in this depth has a maximum influence on e.g. plant growth, water infiltration, and probably also on the storage of OC (Feller and Beare, 1997; Six et al., 2002). Therefore, we used the 0–10 cm topsoil for our study. In the forest soils, we identified two different soil horizons at this depth varying in soil OC and soil structure. Hence, we differentiated between 0–5 cm and 5–10 cm. To have a consistent sampling design, we applied this distinction to the cropland sites, too. We will add this explanation at the end of the introduction. We think that adding more details about the whole experimental approach and methods applied will improve the reader's understanding of the

context (Lines 163–165: "We generally focused on soil samples from 0–10 cm to test our current hypothesis since land-use induced OC losses from soils of the study region largely occur in this depth increment (cf. Kirsten et al., 2019)")

**Referee 2:** "Including a paragraph where it is stated how this manuscript builds up from previous studies would also aid to better highlight the novelty of this study."

**Our response:** We agree with Referee 2 and will prepare a paragraph in the introduction with further information. In addition, we will include an additional sentence in which we indicate that we used the same mineralogical combinations (i.e. soil samples) as in our previous study (Kirsten et al., 2021), but applied a completely different fractionation scheme (Lines 149–154: "In the precursor study, we found a positive relationship between the storage of mineral-associated OC and the ratio of pedogenic Fe to aluminous clay under forest and cropland land use, suggesting that a larger share of Fe oxides is linked to larger OC storage and persistency against land-use change (Kirsten et al., 2021). In the present study, we test whether aggregation and its contribution to OC storage follow similar patterns, or are decoupled from the individual contribution main mineral constituents.", and Lines 159-161: "For this purpose, we determined the aggregate size distribution of soils under both land uses, determined the OC contents of obtained aggregate fractions, and tested the stability of the two largest aggregate size fractions (2–4 mm and > 4 mm).").

**Referee 2:** "The materials and methods also need to include more information regarding the experimental design and sampling protocols. In this version, it is challenging to follow how the database for this study was built, how many samples per plot were taken, and how they were processed. The author references a previous paper (Kirsten et al., 2021) for more details about the experimental design and this paper is also cited in the results section, which is quite confusing. I'm rather unsure of what was measured in this study and what results came from the previously published paper."

**Our response:** We agree with the referee that providing more information about the experimental design would be beneficial for the reader (see response to comments of referee 1). We will add additional information to the material and method section to (i) describe the selection of the study sites, (ii) illustrate that the forests have similar input of fresh organic matter, and (iii) show the similar management practices at the different croplands (Lines 190–197: "The site selection was done based on total clay amount determined in the field and the associated total Fe amount measured with a portable XRF device (Kirsten et al., 2021).

We did not observe systematic differences in vegetation composition of the forest sites and NMR spectra showed a similar composition of litter for each of the two land uses investigated (Kirsten et al., 2021). Furthermore, several visits in the study region over the last decade (2012, 2013, 2015, and 2018) combined with personal communications with farmers and local partners working in the region, enabled us to select cropland sites with similar agricultural management (cultivation of cassava (*Manihot esculenta*), hand hoe tillage, biomass burning before seed bed preparation).").

In the article originally submitted, we already explained how many samples were taken from the six forest and three arable land plots. Nevertheless, we will rephrase the sentence to make this clearer (Lines 197–198: "At each plot, mineral soil from three adjacent and randomly distributed soil pits at mid-slope position was sampled at 0–5 and 5–10 cm depths."). As indicated in our response to referee 1, we will include additional information on sample processing.

**Referee 2:** *"For the discussion, I would suggest the authors be more concrete and build up a stronger argument that links the results with the proposed hypotheses."*

**Our response:** We thank Referee 2 for his comment. We hypothesized that the mineralogical combination resulting in the largest aggregate stability also results in the largest OC persistence under forest and cropland land uses. As indicated in our response to Referee 1, we will clearly point out in the conclusions that we must reject our hypothesis, which is then more clearly deduced from the discussion. We will add to the conclusion section (Lines 598–599: "Hence, we must reject our initial hypothesis that the mineralogical combination that results in the greatest aggregate stability best preserves OC during forests to cropland conversion.").

**Answers to detailed comments of referee 2**

**Abstract**

**Referee 2:** "Lines 30–32: Please split this sentence in two as it is hard to follow the argument."

**Our response:** We agree with referee 2 and will split the sentence in two (Lines 30–32: "However, as aluminosilicates, aluminum oxyhydroxide and Fe oxides are part of the clay-sized fraction it is hard to separate, how certain mineral phases modulate aggregation. In

addition, it is not known what consequences this will have for organic carbon (OC) persistence after land-use change.").

**Referee 2: "Line 34: insert "land uses" after "cropland"**

Our response: We agree with referee 2 and will add "land uses" "after cropland" (Line 34).

**Referee 2:** "Line 41: It is not clear from the statement above about the methods and measurements done, how was the persistence of OC was measured?"

**Our response:** Land-use change from natural forest to cropland is often related to large losses in OC and larger losses indicate smaller OC persistence. We assessed the persistence of OC by comparing OC storage in aggregate fractions between forest and cropland soils for each mineralogical combination. Similar OC storage in the two land uses studied indicates high persistence whereas lower OC values in cropland than in forest soils reflect low persistence. This approach was possible because of very similar environmental conditions and applied to all individual mineralogical combinations with varying contents of aluminous clay and pedogenic Fe oxides. We will add a short description of this approach to the introduction (Lines 161–163: "As a measure of OC persistence, the OC content of aggregate size fractions was compared between the two land uses in the same mineralogical combination.").

**Referee 2:** "*Line 41*: "after the change in land use". Not clear what this means. It makes me think that this study is a Chrono sequence in which impacts on land-use change across time were assessed rather than comparisons done between plots with different land uses."

**Our response:** We agree and will change "after the change in land use" to "even under agricultural use" (Line 43).

**Introduction**

**Referee 2:** "Line 58: "aggregation depends strongly on inorganic cementing agents", like those mentioned in the previous line? Please better link these two sentences if that is the case, or give some examples of the inorganic cementing agents of relevance for the tropics."

**Our response:** We agree with Referee 2 and will rewrite the sentence so that it is better linked to the previous sentence. (Lines 78–80: "The study by Six et al. (2002) points to the special role of inorganic compounds such as clay minerals and pedogenic metal oxides in the formation of aggregates in the tropics.").

Referee 2: "Line 99: remove "into ""

Our response: We agree with Referee 2 and will remove "into" (Line 155).

**Referee 2: "Line 102: "hypothesize"?"**

**Our response:** We agree with Referee 2 and will exchange "presume" with "hypothesize" (Line 158).

**Referee 2:** "Line 103–104: "after conversion of forests into croplands" sounds like it was assessed in a Chrono sequence. Also, I'd suggest splitting this hypothesis in two, one focused on the "mineralogical combination resulting in the largest aggregate stability also results in largest OC persistence" and the other one regarding the impact of land-use change. Are there any hypotheses/predictions related to the depths included as factors? Also, what do you mean by "combination", the proportion of clays vs Fe oxides?"

**Our response:** We agree with Referee 2 and will rephrase the sentence because we did not conduct a chronosequence study (Lines 158–159: "We hypothesize that the mineralogical combination resulting in the largest aggregate stability also results in the largest OC persistence."). Furthermore, we will add additional information how we tried to answer the hypothesis (Lines 159–163: "For this purpose, we determined the aggregate size distribution of soils under both land uses, determined the OC contents of obtained aggregate fractions, and tested the stability of the two largest aggregate size fractions (2–4 mm and > 4 mm). As a measure of OC persistence, the OC content of aggregate size fractions was compared between the two land uses in the same mineralogical combination.").

"Combination" means the proportion of aluminous clay and Fe oxides in a given topsoil. We will clarify that (Lines 154–157: "In detail, our main research goal was to investigate the individual role of aluminous clay and pedogenic Fe oxides for determining...").

We do not have a specific hypothesis related to the two depth increments because we want to keep the focus of the manuscript as straightforward as possible.

**Referee 2:** "Line 103: Here and from the introduction, is still not clear to me what the authors refer to by "OC persistence". This OC property tends to be associated with measurements over time, which makes me wonder again if this study is done in a Chrono sequence of land-use change but I couldn't find enough information in the M&M regarding this."

**Our response:** Please, refer to our reply given above where we explain our approach to determine OC persistence (i.e. comparing OC between forest and cropland soils).

**Referee 2:** "*Line 104–105*: This sentence reads disconnected from the paragraph."**

**Our response:** We agree with Referee 2 and will rephrase the sentence to connect it better to the former sentence (Lines 163–165: "We generally focused on soil samples from 0–10 cm to test our current hypothesis since land-use induced OC losses from soils of the study region largely occur in this depth increment (cf. Kirsten et al. 2019).").

**M&M**

**Referee 2:** "*Line 116*: Could you please provide more information about the characteristics of the soil profiles at the site? I'm guessing 0-5 organic layer and 5-10 is mineral soil?"

**Our response:** We agree with Referee 2 and refer to the additional information given based on the comment of Referee 1. Furthermore, we will clarify that we used only mineral soil (0-5 cm and 5-10 cm) in our study (Lines 197–198: "At each plot, mineral soil from three adjacent and randomly distributed soil pits at mid-slope position was sampled at 0–5 and 5–10 cm depths.").

Detailed information about the soil profiles can be found in the supplementary material of Kirsten et al. (2021) in Section 1: General soil description (Table S1).

**Referee 2:** "*Line 132*: It is not clear how these groups (of what? ...plots? samples?) were determined, please be more specific."

**Our response:** We agree with Referee 2 and will include additional information in which we clarify that each sample is assigned to a specific mineralogical combination based on its respective content of aluminous clay and pedogenic Fe oxide (Lines 224–228: "Based on the respective content of aluminous clay and pedogenic Fe oxide in the 5–10 cm depth increment, each sample was assigned to a certain mineralogical combination. The threshold values for aluminous clay and pedogenic Fe oxides to distinguish between "high" and "low" were set to  $250 \text{ g kg}^{-1}$  and  $60 \text{ g kg}^{-1}$ , respectively.").

**Referee 2:** "*Line 139*: Why is this relevant? Was the soil sampling done during this season?" **Our response:** This information shows that drying at 40°C for sample transport resembles processes under natural conditions and ensures low changes in OC by microbial processes.

**Referee 2:** "*Line 165*: please state here, the n of the experiment and your treatments (landuse and the "mineralogical combination"). Also from results, depth was also a factor? Why?" **Our response:** We showed the number of repetitions in each table and figure. We will add this information to "Statistics and calculations" (Lines 270–273: "Based on our selected threshold values for aluminous clay and pedogenic Fe oxides, we were able to achieve the following number of replicates for the mineralogical combinations: 'low clay–low Fe' under forest (n = 4), 'low clay–high Fe' under forest (n = 4), 'high clay–low Fe' under forest (n = 3), 'high clay–high Fe' under forest (n = 7); all cropland combinations (n = 3).")

**Referee 2:** "*Line 168–169:* Could you be more specific about what were you looking to find with these correlations? What specific hypothesis were you aiming to solve?"

**Our response:** We will rephrase the sentence to be more specific about our aims in applying regression analyses (Lines 263–265: "Regression analysis was used to test for relationships between mineralogical properties and MWD, masses of aggregate size fractions, aggregate stability, and OC losses due to land-use change.").

**Results**

**Referee 2:** "In general, this section needs to be better synthesized and focused only on the results from the present study that are relevant to the proposed hypotheses."

**Our response:** We will try to improve the result section, particularly by clarifying the links to our objectives. We also have checked the presented results regarding their relevance for our objectives and hypothesis. Based on this, we would like to keep parts of section 3.1, summarizing the general soil properties, which are crucial baseline data for this study.

**Referee 2:** "*Line 174–176*: maybe something to include in the introduction instead? This is not part of the results of this study."**

**Our response:** We agree with Referee 2 and will remove this sentence from the results section to the introduction where it will replace a similar sentence (Lines 129–131: "The mineralogical composition of the study soils is very homogeneous with kaolinite and gibbsite as the main aluminous minerals of the clay fraction and goethite and hematite as dominant pedogenic Fe oxides (Kirsten et al., 2021)."). Furthermore, we will add additional information about the consequences for the aluminous clay to Fe oxides ratio in the following sentence

(Lines 131–149: "Yet, the ratio of aluminous clays to Fe oxides differed strongly, giving rise to unique mineralogical combinations under both land use types. Thus, the conversion of natural forest to cropland in the study region...")

**Referee 2:** "*Line 180–181*: *ibid**

Table 1: two soil increments were measured but it's not clear to me why. This table contains a lot of information that is not discussed or mentioned in the text aside from a broad description of the site characteristics. If not that relevant, maybe it belongs to supplements? I would be interested to read a short description of the impact of the treatments: land use, depth, and mineralogical combination on the OC and other variables in this table..."

**Our response:**

Here, we do not agree with Referee 2. In our opinion, Table 1 provides the most important soil properties affecting aggregation and OC and, therefore, we would prefer to keep this table in the main manuscript.

**Referee 2: "Line 192–194: Awkward sentence structure."**

**Our response:** We agree with Referee 2 and will rephrase the sentence (Lines 305–307: "For most combinations, about 74% of soil mass was present in aggregates > 2 mm (Figure 1a), whereas in forest soils with low contents in both aluminous clay and Fe oxides only 40% could be assigned to aggregates > 2 mm. Only 3-12%").

**Referee 2: "Line 228: This analysis was not."**

**Our response:** The model is specified in the supplementary Table S1 under the item "Mass.Aggregates0-10 cm > 2 mm" (aluminous clay = -0.20, pedogenic Fe oxides = 0.85, intercept = 169.81, Df = 51, F-value = 35.95,  $r^2 = 0.57$ , p < 0.01).

**Referee 2:** "*Line 250–257*: This is a great paragraph that really helps to put in context all the above results. Previous paragraphs were too dense so I suggest trying to use more of this sort of narrative to describe the results of the study, given all the variables analyzed."

**Our response:** This is exactly what we wanted to achieve – a short and concise summary of the results related to the observed aggregate size distribution. The other two sections of the results are less dense and less diverse. Therefore, we do not think that similar summaries would be necessary / helpful for the reader.

**Referee 2:** "*Line 287 and 291*: Please focus on the results of your study, this is an example of when it is not clear what was done in this study vs the author's previous publication."

**Our response:** We will remove these two sentences in order to make crystal clear the results from the current study in comparison to the previous one.

Referee 2: "Line 300–301: Not fully certain what is the support for this statement."

**Our response:** We agree with Referee 2. We will remove the sentence, because this aspect is already included in the discussion (Line 427).

**Discussion**

Referee 2: "Line 316: respectively? Which value belongs to what? Not clear."

**Our response:** We agree with Referee 2 and will restructure and rephrase this sentence (Lines 446–452: "For example, soils with strongly contrasting clay content (220 and 650 g kg-1) but similar clay mineralogy (kaolinite) in the central highlands of Kenya displayed macroaggregate contents of 245 and 636 g kg-1 soil, respectively (Gentile et al., 2010). In addition, for soils from the catchment of the Riru river also located in the central highlands of Kenya it was shown that macroaggregates (2–4.2 mm) displayed a large stability (Kamamia et al., 2021). The reported MWD's after application of the fast-wetting stability test were 2.5 mm for cropland and 3.2 mm for indigenous forest sites (Kamamia et al., 2021).").

Referee 2: "*Line 335–338*: *Neat*!" Our response: Thanks.

**Referee 2:** "*Line 357*: replace "elsewhere" to like reported in Rabbi et al., 2015 (without the parenthesis)."

Our response: We agree with Referee 2 and will remove the word "elsewhere" (Line: 517).

**Referee 2:** "*Line 358–359*: I was not under the impression, from the introduction, that confirming this was the purpose of this study."

**Our response:** The aim of our study was not to determine the role of pedogenic Fe oxides for macroaggregation in general. Our focus was to investigate the individual role of aluminous clays and pedogenic Fe oxides (Lines 154–155) on the soil aggregate size distribution. Therefore, we chose to open this paragraph with the excellent study by Six et al. (2002), and further showed that our results differ from the findings given there.

**Referee 2: "Line 398: taken into account... in the models?"**

**Our response:** We refer in this sentence on data given in Table 2 and the significant differences observed between forest and cropland. We will rephrase the sentence and will add this information to the sentence. (Lines 560–563: "Comparing forest with cropland soils (Table 2), we observed significantly reduced OC contents in the majority of macroaggregate fractions of the low clay–low Fe and high clay–high Fe croplands, as reported in other studies (Blanco-Canqui and Lal, 2004; Lobe et al., 2011).").

**Referee 2: "Line 401: fewer changes"**

**Our response:** We agree with Referee 2 and will replaced the word "least" with "fewer" (Line 563).

**Referee 2:** "Line 408–412: This is for example a way in which the fractionation and aggregate characterization did in this study build up from the previous paper by the authors." **Our response:** Thank you for this comment. We tried to improve this in many parts of the manuscript.

We would like to thank both reviewers for their meaningful and constructive comments, which were really helpful to improve the entire manuscript. We would also like to thank the editorial board for giving us the opportunity to improve our manuscript.

Sincerely yours,

M. Kinh

Maximilian Kirsten

**References**

- Barthès, B. G., Kouakoua, E., Larré-Larrouy, M.-C., Razafimbelo, T. M., de Luca, Edgar F., Azontonde, A., Neves, C. S.V.J., de Freitas, Pedro L., and Feller, C. L.: Texture and sesquioxide effects on water-stable aggregates and organic matter in some tropical soils, Geoderma, 143, 14–25, https://doi.org/10.1016/j.geoderma.2007.10.003, 2008.
- Blanco-Canqui, H. and Lal, R.: Mechanisms of Carbon Sequestration in Soil Aggregates, CRC Crit. Rev. Plant Sci., 23, 481–504, https://doi.org/10.1080/07352680490886842, 2004.
- Coward, E. K., Thompson, A. T., and Plante, A. F.: Iron-mediated mineralogical control of organic matter accumulation in tropical soils, Geoderma, 306, 206–216, https://doi.org/10.1016/j.geoderma.2017.07.026, 2017.
- Don, A., Schumacher, J., and Freibauer, A.: Impact of tropical land-use change on soil organic carbon stocks a meta-analysis, Glob. Change Biol., 17, 1658–1670, https://doi.org/10.1111/j.1365-2486.2010.02336.x, 2011.
- Dultz, S., Woche, S. K., Mikutta, R., Schrapel, M., and Guggenberger, G.: Size and charge constraints in microaggregation: Model experiments with mineral particle size fractions, Applied Clay Science, 170, 29– 40, https://doi.org/10.1016/j.clay.2019.01.002, 2019.
- Feller, C. and Beare, M. H.: Physical control of soil organic matter dynamics in the tropics, Geoderma, 79, 69–116, https://doi.org/10.1016/S0016-7061(97)00039-6, 1997.
- Gee, G.W. and Bauder, J.W.: Particle-size analysis, in: Methods of soil analysis: Part 1 Physical and mineralogical methods, 2nd ed. // 2nd ed, edited by: Klute, A. and Page, A. L., American Society of Agronomy; Soil Science Society of America, Madison, 383–412, 1986.
- Gentile, R., Vanlauwe, B., Kavoo, A., Chivenge, P., and Six, J.: Residue quality and N fertilizer do not influence aggregate stabilization of C and N in two tropical soils with contrasting texture, Nutr. Cycling Agroecosyst. (Nutrient Cycling in Agroecosystems), 88, 121–131, https://doi.org/10.1007/s10705-008-9216-9, 2010.
- Hou, T., Xu, R., and Zhao, A.: Interaction between electric double layers of kaolinite and Fe/Al oxides in suspensions, Colloids and Surfaces A: Physicochemical and Engineering Aspects, 297, 91–94, https://doi.org/10.1016/j.colsurfa.2006.10.029, 2007.
- John, B., Yamashita, T., Ludwig, B., and Flessa, H.: Storage of organic carbon in aggregate and density fractions of silty soils under different types of land use, Geoderma, 128, 63–79, https://doi.org/10.1016/j.geoderma.2004.12.013, 2005.
- Kamamia, A. W., Vogel, C., Mwangi, H. M., Feger, K.-H., and Julich, S.: Mapping soil aggregate stability using digital soil mapping: A case study of Ruiru reservoir catchment, Kenya, Geoderma Regional, 24, 2021.
- Kirsten, M., Mikutta, R., Vogel, C., Thompson, A., Mueller, C. W., Kimaro, D. N., Bergsma, H. L. T., Feger, K.-H., and Kalbitz, K.: Iron oxides and aluminous clays selectively control soil carbon storage and stability in the humid tropics, Scientific Reports, 11, https://doi.org/10.1038/s41598-021-84777-7, 2021.
- Kirsten, M., Kimaro, D. N., Feger, K.-H., and Kalbitz, K.: Impact of land use on soil organic carbon stocks in the humid tropics of NE Tanzania, J. Plant Nutr. Soil Sci., 182, 625–636, https://doi.org/10.1002/jpln.201800595, 2019.
- Kleber, M., Eusterhues, K., Keiluweit, M., Mikutta, C., Mikutta, R., and Nico, P. S.: Mineral–Organic Associations: Formation, Properties, and Relevance in Soil Environments, Adv. Agron., 130, 1–140, https://doi.org/10.1016/bs.agron.2014.10.005, 2015.
- Lawrence, C. R., Harden, J. W., Xu, X., Schulz, M. S., and Trumbore, S. E.: Long-term controls on soil organic carbon with depth and time: A case study from the Cowlitz River Chronosequence, WA USA, Geoderma, 247-248, 73–87, https://doi.org/10.1016/j.geoderma.2015.02.005, 2015.
- Lehmann, J. and Kleber, M.: The contentious nature of soil organic matter, Nature, 113, 143, https://doi.org/10.1038/nature16069, 2015.
- Lobe, I., Sandhage-Hofmann, A., Brodowski, S., du Preez, C. C., and Amelung, W.: Aggregate dynamics and associated soil organic matter contents as influenced by prolonged arable cropping in the South African Highveld, Geoderma, 162, 251–259, https://doi.org/10.1016/j.geoderma.2011.02.001, 2011.

- Maltoni, K. L., Mello, L. M. M. de, and Dubbin, W. E.: The effect of Ferralsol mineralogy on the distribution of organic C across aggregate size fractions under native vegetation and no-tillage agriculture, Soil Use Manag., 33, 328–338, https://doi.org/10.1111/sum.12339, 2017.
- Mehra, O. P. and Jackson, M. L.: Iron Oxide Removal from Soils and Clays by a Dithionite-Citrate System Buffered with Sodium Bicarbonate, Clays Clay Miner., 7, 317–327, https://doi.org/10.1346/CCMN.1958.0070122, 1958.
- Mikutta, R., Turner, S., Schippers, A., Gentsch, N., Meyer-Stüve, S., Condron, L. M., Peltzer, D. A., Richardson, S. J., Eger, A., Hempel, G., Kaiser, K., Klotzbücher, T., and Guggenberger, G.: Microbial and abiotic controls on mineral-associated organic matter in soil profiles along an ecosystem gradient, Sci. Rep., 9, 10294, https://doi.org/10.1038/s41598-019-46501-4, 2019.
- Qafoku, N. P. and Sumner, M. E.: Adsorption and Desorption of Indifferent Ions in Variable Charge Subsoils, Soil Science Society of America Journal, 66, 1231–1239, https://doi.org/10.2136/sssaj2002.1231, 2002.
- Schmidt, M. W. I., Torn, M. S., Abiven, S., Dittmar, T., Guggenberger, G., Janssens, I. A., Kleber, M., Kögel-Knabner, I., Lehmann, J., Manning, David A C, Nannipieri, P., Rasse, D. P., Weiner, S., and Trumbore, S. E.: Persistence of soil organic matter as an ecosystem property, Nature, 478, 49–56, https://doi.org/10.1038/nature10386, 2011.
- Six, J., Feller, C., Denef, K., Ogle, S. M., Moraes, J. C. de, and Albrecht, A.: Soil organic matter, biota and aggregation in temperate and tropical soils Effects of no-tillage, Agronomie, 22, 755–775, https://doi.org/10.1051/agro:2002043, 2002.
- Totsche, K. U., Amelung, W., Gerzabek, M. H., Guggenberger, G., Klumpp, E., Knief, C., Lehndorff, E., Mikutta, R., Peth, S., Prechtel, A., Ray, N., and Kögel-Knabner, I.: Microaggregates in soils, J. Plant Nutr. Soil Sci., 181, 104–136, https://doi.org/10.1002/jpln.201600451, 2018.

---

## Author Response (AR2)

Dresden University of Technology – Dept. Soil Science and Site Ecology
Pienner Straße 19 • 01737 Tharandt, Germany

To the Editorial Board

Copernicus Publications

Journal Soil

Dear Editorial Team,

Thank you for giving us the opportunity to do the minor revision. We found the comments and questions by the unknown referee #2 very useful and suggest revisions of the manuscript accordingly. All of our responses are listed below.

**Response letter**

Response to the comments and questions of the referee's related to the manuscript:

**MS No.: soil-2020-98**

Special Issue: Tropical biogeochemistry of soils in the Congo Basin and the African Great Lakes region

With the title:

"**Aluminous clay and pedogenic Fe oxides modulate aggregation and related carbon contents in soils of the humid tropics**"

**Response to the comments of referee 2.**

**Referee 2:** "*I think the comments are easy to address and I would rate them as a minor revision even though the changes in text for section 3.2 and 4.1 might still be more substantial. While revising sections 3.2 and 4.1, consider that SOIL has a very broad readership and I think it is worthwhile going the extra length in revising and streamlining the MS (especially section 4.1).*"

**General response:** We will revise the logical structure of sections 3.2 and 4.1 as suggested by reviewer 2. We will further try to focus on the core messages of the article in order to improve the readability of the article in these sections.

**Introduction:**

**Referee 2:** "*Line 72: Not sure what other sources of uncertainty the authors are listing here as the previous sentences were examples of the relevance of individual cementing agents on aggregation. Please rephrase this sentence and better connect the argument starting from line 68 up to line 80 to enhance clarity about your problem statement.*"

**Our response:** With this sentence we want to summarize the fact that different authors have recognized various influencing factors on the aggregation and we also assume that a different relative composition of the clay fraction (aluminous clay and pedogenic Fe) may explain these differences. To make our intention clearer, we have rephrased the sentence (Lines 72–75: "Such kind of uncertainty may derive from the fact that the clay size particle fraction (< 2-µm) not only contains OM and different types of clay minerals, but also variable contents of pedogenic Fe and aluminum (Al) oxides (*Barré* et al. 2014; *Fernández-Ugalde* et al. 2013; *Wagai* and *Mayer* 2007)."`.

**Referee 2:** "*Line 81: This was not really stated above. Consider starting the paragraph with "Soil aggregation..."*"

**Our response:** We agree with referee 2 and will remove "As indicated above,..." to start the sentence with "Soil aggregation..." (Line: 83).

**Referee 2:** "*Line 108: insert "of" main mineral constituents*"

**Our response:** We agree with referee 2 and will add "of...main mineral constituents" (Line: 111).

**Referee 2:** "*Line 117: Good to add this clarification but neither here, nor in the methods the authors include an explanation as per why this depth increment was further separated from 0-5 and 5 to 10 as explained in the response to reviewers "In the forest soils, we identified two different soil horizons at this depth varying in soil OC and soil structure. Hence, we differentiated between 0–5 cm and 5–10 cm. To have a consistent sampling design, we applied this distinction to the cropland sites, too" Please add this somewhere in the methods, line 136 perhaps?*"

**Our response:** We agree with referee 2 and will add additional information (Lines 139–142: "This procedure was chosen because we identified two soil horizons at 0–5 and 5–10 cm depth based on differences in color and structure. To have a consistent sampling design, we applied this distinction to the cropland sites, too."

**Methods:**

**Referee 2:** "*Line 193: change treatments for "land uses and depths"*"

**Our response:** We agree with referee 2 and will rephrase the sentence (Lines 198–201: "To test for significant differences between mineralogical combinations, land uses, and depths, we applied the linear model function [lm()] in combination with analysis of variance [aov(lm()].").

**Referee 2:** "*Line 201: how can the n for 'high clay–high Fe' under forest be 7 if only 6 plots from forests were used?*"

**Our response:** This is based on the fact that we have drawn clear threshold values for the content of aluminous clay and pedogenic Fe for assignment of samples to defined mineralogical combinations. As a result, the six forest locations examined were distributed over the four selected mineralogical combinations and caused a different number of repetitions in each combination.

**Results:**

**Referee 2:** "*Table 1: What do the capital letters in Fe$_d$/clay ratio represent?*"

**Our response:** We understand the resulting ambiguity and will adjust the text of the table and remove the capital letters for the Fe$_d$/clay ratio. We'll add the following sentence to clarify what the capital letters mean for the OC content (Lines 224–225: "Lower case letters

indicate significant differences within a certain land use as separated by depth, and capital letters denote significant differences between land uses.").

**Referee 2:** "*Section 3.2: This section is still convoluted and hard to follow. To enhance the readability of this section, maybe focus first on the results from Figure 1 to then describe results from Table 2 or whichever order the authors consider best.*"

**Our response:** We agree with referee 2 and will streamline section 3.2. For this purpose we will remove the following sentences to bring focus on the most relevant results of the manuscript (Lines 243–...: "For most combinations, about 74% of soil mass was present in aggregates > 2 mm (Figure 1a), whereas in forest soils with low contents in both aluminous clay and Fe oxides only 40% could be assigned to aggregates > 2 mm. Only –12% of total soil mass remained in < 0.25 mm aggregates (Table 2)."; Lines 304–...:"The same model separated by soil depth showed similar relationships (Table S1)."; Lines 319–...:" Only under cropland we observed a negative effect of aluminous clay and a positive influence of Fed on microaggregate contents (aggregate mass < 0.25 mm$_{0–5 cm}$: $r^2 = 0.8$, $p = 0.004$; aggregate mass < 0.25 mm$_{5–10 cm}$: $r^2 = 0.61$, $p = 0.03$).")

Furthermore, we will rephrase the section header and will add subheadings to guide the reader through the complex matter of the section (Line 240: "3.2 Influence of aluminous clay and pedogenic Fe on aggregate size distribution"; Line 241: "*Mean weight diameter*"; Line 256: "*Macroaggregate s > 4 mm and 2–4 mm*"; Line 306: "*Microaggregates < 0.25 mm*"; Line 335: "*Summary*").

We will take information out of the section and refer to the corresponding tables to facilitate reading (Line 244: "and 3.7 mm in 5–10 cm depth ("; Line 245: "and 3.7 mm in 5–10 cm depth (, and 4.6 mm in 5–10 cm depth"; Line 247: "MWD$_{Forest\ 5–10\ cm}$: $r^2 = 0.15$, $p = 0.06$"; Line 248: "(MWD$_{Forest\ 0–5\ cm}$: $r^2 < 0.01$, $p = 0.79$; MWD$_{Forest\ 5–10\ cm}$: $r^2 < 0.01$, $p = 0.30$, Table S1)"; Line 250: "0–5 cm depth and 2.7 mm in 5–10 cm").

.

**Referee 2:** "*Line 256: Figure 1a does not report significance results, please add the table where these results are.*"

**Our response:** We do not agree with referee 2, because figure 1a was introduced to the manuscript to provide the reader a fast overview about the general trend in the aggregate mass distribution, which means that the means of both depth increments calculated to reduce complexity of the data. The detailed results, differentiated between the 0–5 and 5–10 cm

depths with their significant differences are given in table 2 already. Adding another table to the manuscript would distract the reader from the main results of the manuscript.

**Referee 2:** "*Line 316: across all plots(?), including both land uses and depths*"

**Our response:** We agree with referee 2 and will rephrase the sentence (Lines 373–374: " In the entire data set, variation in mineral constituents caused pronounced differences in the OC content of the soils between 19 to 95 g OC kg$^{-1}$ (Table 1).").

**Discussion:**

**Referee 2:** "*Section 4.1: I suggest to better synthesize this section. Please focus only on the main results and observed trends that allow you to support/reject your hypothesis. Maybe split this section and discuss separately the impact of land-use change on aggregation and aggregate stability.*"

**Our response:** We agree with Referee 2 and will introduce subheadings to clarify what focus we are covering in the relevant paragraphs (Line 420:" *Mineralogical control on the formation of large macroaggregates*"; Lines 491:" *Land use impact on aggregation within mineral combinations – implications for aggregate stability*"). We try to discuss our objectives point by point in order to be able to answer our hypothesis, but due to the complex nature of the aggregation, a certain intersection of our results is essential. Nonetheless, we will change the order of the paragraphs to first discuss the effects of aluminous clay and pedogenic Fe on aggregation with focus on macroaggregation, followed by the effects of land use change.

We will further reorganize the following sentence thematically and shorten it (Lines 445 –...: "Consequently, this rather indicates the importance of kaolinite for macroaggregation, which is in line with the results of two oxisols in Brazil (Vrdoljak and Sposito, 2002) kaolinite is the backbone of the aggregate size fractions examined. "), and will add the sentence to the previous paragraph (Lines 439–441: "This is in line with results from two Oxisols in Brazil (Vrdoljak and Sposito, 2002), showing kaolinite being the backbone of macroaggregates.").

**Referee 2:** "*Line 442: "does" not*"

**Our response:** We agree with referee 2 and will add "does" to the sentence (Line: 538).

**Referee 2:** "*Line 467: Please abstain from using words that imply a temporal dimension like "during" when comparing forest and croplands, it gives the false impression that the study included a temporal scale.*"

**Our response:** We agree with referee 2 and will rephrase the sentence (Lines 563–565: "Despite the high physical stability, OC contents of macroaggregates declined substantially in most mineralogical combinations if forest was compared with cropland land use.").

We would like to thank referee 2 for the meaningful and constructive comments again, which were really helpful to further improve the entire manuscript. We would also like to thank the editorial board for giving us the opportunity to improve our manuscript again.

Sincerely yours,

Maximilian Kirsten

---

## Author Response (AR3)

Dresden University of Technology – Dept. Soil Science and Site Ecology
Pienner Straße 19 • 01737 Tharandt, Germany

To the Editorial Board

Copernicus Publications

Journal Soil

Dear Editorial Team,

Thank you for the opportunity to publish our research results in SOIL. We will make the technical changes to improve the manuscript. We listed all of our responses below.

**Response letter**

Response to the technical corrections requested by the executive editor:

**MS No.: soil-2020-98**

Special Issue: Tropical biogeochemistry of soils in the Congo Basin and the African Great Lakes region

With the title:

"**Aluminous clay and pedogenic Fe oxides modulate aggregation and related carbon contents in soils of the humid tropics**"

**Response to the executive editor**

**Executive editor (1):** "*Referee 2: "Line 201: how can the n for 'high clay–high Fe' under forest be 7 if only 6 plots from forests were used?"*

*Please make sure that your answer this comment is also implemented in its content in the paper. Make it clear for the reader where (and why) the number of observations differ in certain parts of your work. From your answer, I am not sure if you have implemented this already in the MS itself.*"

**Response:** To make it clear to the reader that the distribution of the six forests examined in four mineralogical combinations led to an unequal number of repetitions, we will add additional information in the method section. (_Lines 162–164_: "Using the threshold criterion for assigning the individual samples to a mineralogical combination resulted in an unequal number of repetitions for mineralogical combinations under forests ($n = 3$-$7$) whereas those under cropland remained the same ($n = 3$).").

**Executive editor (2):** "*in line 335 you start now with a "summary". I find this a bit odd since you are in the middle of your results section still. I think your original way of writing "in summary" without creating a new subheader was better. But if you think this breaks the style of section 3.2 maybe you can find a way to implement those sentences in the paragraphs before to which they relate to.*"

**Response:** We will replace the subheading "summary" with the running text summary of this paragraph (_Lines 276–278_: "In summary, mineralogical combinations and land use significantly affected the aggregate size distribution of soils, despite quantitative relations to mineralogical proxies could not be observed for each aggregate class.").

We would like to thank the executive editor for the constructive comments. We would also like to thank the entire editorial team for always being at our side during the publication process.

Sincerely yours,

Maximilian Kirsten